# Novel analytical tools reveal that local synchronization of cilia coincides with tissue-scale metachronal waves in zebrafish multiciliated epithelia

Christa Ringers[1,2,3], Stephan Bialonski[4,5], Mert Ege[1], Anton Solovev[5,6], Jan Niklas Hansen[2], Inyoung Jeong[1], Benjamin M Friedrich[5,6]*, Nathalie Jurisch-Yaksi[1,2]*

[1]Department of Clinical and Molecular Medicine, Norwegian University of Science and Technology, Trondheim, Norway; [2]Kavli Institute for Systems, Neuroscience and Centre for Neural Computation, Norwegian University of Science and Technology, Trondheim, Norway; [3]Department of Pharmaceutical Biosciences and Science for Life Laboratory, Uppsala University, Uppsala, Sweden; [4]Institute for Data-Driven Technologies, Aachen University of Applied Sciences, Jülich, Germany; [5]Center for Advancing Electronics, Technical University Dresden, Dresden, Germany; [6]Cluster of Excellence 'Physics of Life', Technical University Dresden, Dresden, Germany

*For correspondence:
benjamin.m.friedrich@tu-dresden.de (BMF);
nathalie.jurisch-yaksi@ntnu.no (NJ-Y)

Competing interest: The authors declare that no competing interests exist.

**Abstract** Motile cilia are hair-like cell extensions that beat periodically to generate fluid flow along various epithelial tissues within the body. In dense multiciliated carpets, cilia were shown to exhibit a remarkable coordination of their beat in the form of traveling metachronal waves, a phenomenon which supposedly enhances fluid transport. Yet, how cilia coordinate their regular beat in multiciliated epithelia to move fluids remains insufficiently understood, particularly due to lack of rigorous quantification. We combine experiments, novel analysis tools, and theory to address this knowledge gap. To investigate collective dynamics of cilia, we studied zebrafish multiciliated epithelia in the nose and the brain. We focused mainly on the zebrafish nose, due to its conserved properties with other ciliated tissues and its superior accessibility for non-invasive imaging. We revealed that cilia are synchronized only locally and that the size of local synchronization domains increases with the viscosity of the surrounding medium. Even though synchronization is local only, we observed global patterns of traveling metachronal waves across the zebrafish multiciliated epithelium. Intriguingly, these global wave direction patterns are conserved across individual fish, but different for left and right noses, unveiling a chiral asymmetry of metachronal coordination. To understand the implications of synchronization for fluid pumping, we used a computational model of a regular array of cilia. We found that local metachronal synchronization prevents steric collisions, i.e., cilia colliding with each other, and improves fluid pumping in dense cilia carpets, but hardly affects the direction of fluid flow. In conclusion, we show that local synchronization together with tissue-scale cilia alignment coincide and generate metachronal wave patterns in multiciliated epithelia, which enhance their physiological function of fluid pumping.

## Editor's evaluation

This fundamental study reports new observations on the coordination of cilia in zebrafish multiciliated epithelia. The work combines novel experimental methods and computation to provide convincing evidence for a conjectured relationship between local and global synchronization in the

form of metachronal waves. The work will be of broad interest to researchers in the areas of cell biology, development, and physiology.

## Introduction

Motile cilia are highly conserved, hair-like cell appendages that beat periodically to move fluid in a wide range of species. In vertebrates, motile cilia are present in the brain, the respiratory system, reproductive tracts, and the left-right organizer, where they serve numerous functions related to fluid transport. Cilia move cerebrospinal fluid along the ventricles of the vertebrate brain (*Olstad et al., 2019*; *Sawamoto et al., 2006*; *Ringers et al., 2020*; *Faubel et al., 2016*; *Worthington and Cathcart, 1966*, *Date et al., 2019*; *D'Gama et al., 2021*) and spinal cord (*Sternberg et al., 2018*; *Thouvenin et al., 2020*), contribute to mucociliary clearance, which protect our lungs and nose from pathogens (*Bustamante-Marin and Ostrowski, 2017*; *Ramirez-San Juan et al., 2020*, *Wallmeier et al., 2019*), or establish the left-right body axis during embryonic development (*Ferreira et al., 2018*; *Nonaka et al., 1998*). Cilia often coordinate their movements within and across cell boundaries (*Sanderson and Sleigh, 1981*; *Brumley et al., 2012*; *Machemer, 1972*; *Knight-Jones, 1954*), which may improve cilia-mediated fluid transport as suggested by modeling studies (*Elgeti and Gompper, 2013*). Yet, how cilia coordination emerges in multiciliated tissues remains an active topic of research.

In the absence of a tissue-scale ciliary pacemaker, cilia require a physical coupling to coordinate their beat. Coupling between cilia can occur through the surrounding fluid, which is referred to as hydrodynamic coupling, as originally proposed by *Taylor, 1952* and first demonstrated experimentally for pairs of cilia (*Brumley et al., 2014*). In short, the movement of one cilium sets the surrounding fluid in motion, which impacts hydrodynamic friction forces on other cilia in its vicinity. A change in hydrodynamic friction forces causes cilia to beat slightly faster or slower (*Klindt et al., 2016*). Theoretical studies showed that hydrodynamic interactions could, at least in principle, orchestrate self-organized synchronization of cilia in metachronal waves, which refer to a sequential rather than synchronous or random movement of neighboring cilia (*Gueron and Levit-Gurevich, 1999*; *Guirao and Joanny, 2007*; *Osterman and Vilfan, 2011*; *Wollin and Stark, 2011*; *Elgeti and Gompper, 2013*; *Meng et al., 2021*; *Solovev and Friedrich, 2022b*; *Chakrabarti et al., 2022*; *Kanale et al., 2022*). Additionally, basal coupling of cilia through their anchoring in the cell cortex may contribute to cilia synchronization (*Quaranta et al., 2015*; *Wan and Goldstein, 2016*; *Klindt et al., 2017*; *Soh et al., 2020*). Putative synchronization mechanisms must be sufficiently strong to overcome the deleterious effects of slightly distinct beat frequencies, active cilia noise (*Polin et al., 2009*; *Ma et al., 2014*; *Goldstein et al., 2009*; *Solovev and Friedrich, 2022a*), and disorder in cilia alignment (*Guirao et al., 2010*). Hence, global synchronization across an entire tissue is rather unlikely.

The localization of cilia carpets deep inside the vertebrate body has motivated previous studies of cilia carpet dynamics to resort to more accessible and simpler, non-animal model systems on the surface of protists such as *Paramecium* (*Machemer, 1972*) or the green alga *Volvox* (*Machemer, 1972*; *Brumley et al., 2012*), or cell culture systems (*Pellicciotta et al., 2020*; *Oltean et al., 2018*; *Gsell et al., 2020*; *Khelloufi et al., 2018*). Intriguingly, cilia coordination in the form of metachronal waves is observed in some systems, but not in others, prompting the question on the determinants of cilia synchronization. Previous work showed that cilia density and spatial distribution (*Pellicciotta et al., 2020*; *Khelloufi et al., 2018*), the viscosity of the surrounding fluid (e.g. mucus versus water) (*Machemer, 1972*; *Gheber et al., 1998*), as well as the tissue-scale alignment of cilia polarity (*Mitchell et al., 2009*) appear to be key parameters for synchronization in cilia carpets. Intriguingly, the wave direction of these metachronal waves can be rotated relative to the direction of the effective stroke of the cilia beat (*Knight-Jones, 1954*). This broken chiral symmetry of metachronal waves is likely a consequence of chiral cilia beat patterns (*Elgeti and Gompper, 2013*; *Solovev and Friedrich, 2022b*; *Meng et al., 2021*), but the details are not understood. Finally, it is not even clear to what extent metachronal coordination is rather a local or a tissue-scale phenomenon, and how this in turn affects the pumping rate of cilia carpets. Answering these open questions requires accessible model systems.

In this study, we combined experimental and theoretical work to study the coordination of motile cilia and its impact on fluid pumping. We chose as main experimental model the larval zebrafish nose, a cup-like structure of approximately 50 μm diameter containing many multiciliated cells (*Reiten et al., 2017*), which allows for non-invasive biophysical measurements of cilia beating due to its

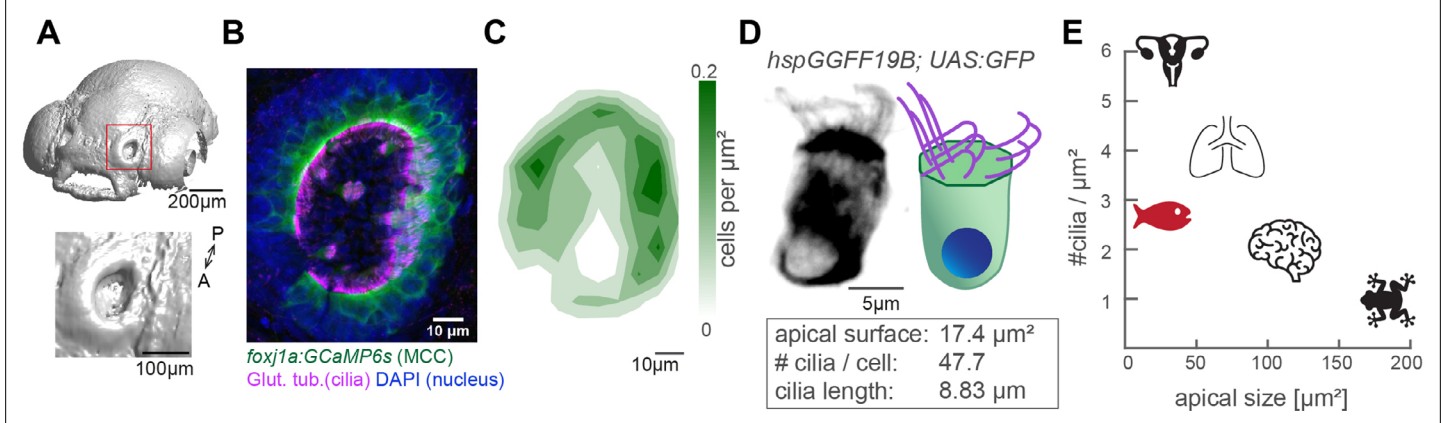

**Figure 1.** The zebrafish nose as model system for a ciliated epithelium with small and densely packed multiciliated cells. (**A**) Surface rendering of a 4-day-old zebrafish larva (top) and a zoom-in of the nasal cavity (bottom). (**B**) A representative example of a left nose marked by a red box in (**A**). In the maximum projection, motile cilia are labeled in magenta (glutamylated tubulin), nuclei in blue (DAPI), and multiciliated cells in green (*foxj1a:GCaMP6s*). Note the lack of multiciliated cells in the center of the nose. DAPI signals highlight the presence of other cell types. (**C**) A contour plot showing the average multiciliated cell density (maximum projection) with a total number of 50.8 multiciliated cells per fish (±6.2 SD; n=15). (**D**) A representative example (left) and schematic (right) of a multiciliated cell labelled in the transgenic line *hspGGFF19B:UAS:GFP*. On average, each cell has 47.7 cilia (±9.9 SD; n=4), the apical surface spans 17.4 µm² (±6.3 SD; n=11), and cilia are 8.83 µm long (±0.86 SD; n=38; *Figure 1—figure supplement 1B-E'*). (**E**) A graph depicting ciliary density per cell across animals and organs. Shown are the zebrafish nose, clawed frog skin (*Klos Dehring et al., 2013*; *Kulkarni et al., 2021*), mouse brain ventricles (*Redmond et al., 2019*), lungs (*Nanjundappa et al., 2019*), and oviduct (*Shi et al., 2014*). All n refer to the number of fish. SD = standard deviation, A: anterior, P: posterior.

The online version of this article includes the following figure supplement(s) for figure 1:

**Figure supplement 1.** Quantification of multiciliated cell features in the zebrafish nose.

superficial location. Using this in vivo model, we observed that the cilia beat frequency is heterogeneous across the epithelium, limiting synchronization to local domains. Notwithstanding, we identified robust metachronal coordination, with waves that propagated along stable directions across the multiciliated epithelium. The direction of metachronal waves is consistently different in the right nose as compared to the left nose. This difference between left and right noses marks an instance of broken chiral symmetry, which is not explained by cilia (*Gheber et al., 1998*) orientation. To understand the implications for fluid pumping of local synchronization as observed in the zebrafish nose, we used a computational model of a regular array of cilia (*Solovev and Friedrich, 2022b*; *Solovev and Friedrich, 2022a*). We detail how local synchronization is necessary to avoid steric collisions between cilia at higher cilia densities, which helps efficient cilia beating. At the same time, the rate of fluid pumping is robust against weak levels of noise. Finally, we showed that the direction and wavelength of metachronal traveling waves influence the mean pumping rate, but hardly affects the direction of fluid flow.

Altogether, we propose that local synchronization, tissue-scale cilia alignment, and the emergence of global metachronal waves coincide in a cilia carpet. Collectively, they support its physiological function of fluid pumping.

## Results

### The dense packing of multiciliated cells in the zebrafish nose provides a powerful model to study cilia coordination

To study how cilia beating coordinates throughout an entire ciliated organ in its native state, we turned to the larval zebrafish nose, as its localization on the snout allows for non-invasive live imaging of an entire tissue (*Figure 1A* & *Figure 1—figure supplement 1A*). At 4-day post fertilization, the zebrafish nose is a cup-like structure containing multiciliated cells at its rim and olfactory sensory neurons and support cells in its center (*Kermen et al., 2013*; *Reiten et al., 2017*; *Hansen and Zeiske, 1993*). To characterize the three-dimensional organization of motile cilia in this organ, we quantified the number, distribution, and properties of multiciliated cells. We imaged 4-day-old larvae that expressed a GFP-based indicator within multiciliated cells (*Tg(Foxj1a:GCaMP6s)*), and that were stained for the

ciliary markers, acetylated tubulin (*Reiten et al., 2017*) and glutamylated tubulin (*Olstad et al., 2019*; *Pathak et al., 2014*, *Figure 1B* and *Figure 1—figure supplement 1B*). Our results revealed that the distribution of multiciliated cells across the 3D geometry of the nose is stereotypical: most cells populate the lateral rim, forming two to three rows of multiciliated cells, while fewer cells occupy the medial rim, and are even absent in the anteromedial center regions (*Figure 1B&C*). This pattern is highly consistent across fish and mirrored for left and right noses (*Figure 1—figure supplement 1B*). We identified that on average each nose contains 50.8 multiciliated cells (±6.2; *Figure 1—figure supplement 1B–B'*). We also measured that 47.7 cilia (±9.9; *Figure 1—figure supplement 1C–C'*) emanate from an apical cell surface of 17.4 μm² (±6.3; *Figure 1—figure supplement 1D–D'*), with a length of 8.8 μm (±0.9; *Figure 1—figure supplement 1E–E'*). From the abovementioned values (*Figure 1D*), we calculated the distance between cilia to be 0.68 μm, and an area density of 2.7 cilia/μm². Altogether, our results revealed that multiciliated cells in the four-day old zebrafish nose contain fewer cilia than multiciliated cells of the lung epithelium (>100 cilia per cell) (*Nanjundappa et al., 2019*) or *Xenopus laevis* embryonic skin (~150 cilia per cell) (*Klos Dehring et al., 2013*; *Kulkarni et al., 2021*), but retain a similar cilia density due to their reduced apical surface (*Figure 1E*). Because of this, we argue that the larval zebrafish nose is a powerful model to study cilia coordination within and across neighboring cells in a three-dimensional multiciliated carpet.

## Frequency heterogeneity restricts the synchronization of cilia to a local scale

To understand the dynamic properties of cilia beating in our system and their impact on ciliary coordination, we recorded cilia beating with light transmission microscopy and measured the ciliary beat frequency (CBF) using a Fast Fourier Transform (FFT)-based method (*Figure 2A–C* and Materials and methods) adapted from *Reiten et al., 2017*. Using this technique, we observed that CBF is heterogeneous in the nose but organized in frequency patches (*Figure 2C* and *Figure 2—figure supplement 1A*). This CBF map remained stable over time (*Figure 2—figure supplement 1B*) and across different depths of the recording (*Figure 2—figure supplement 1C*). To investigate the origin of frequency patches, we tested whether individual patches correspond to the beating of individual cells. Specifically, we examined cilia beating in transgenic animals expressing GFP sparsely in multiciliated cells of the nose (*Et(hspGGFF19B:Gal4)Tg(UAS:gfp)* *Reiten et al., 2017*). We then recorded cilia beating sequentially with light-sheet microscopy, where we can identify individual cells, and light-transmission microscopy, where we can record the entire cilia carpet, and compared their CBF maps. We observed that the CBF of an individual cell corresponds better to that of the frequency patch than to the rest of the cilia carpet (*Figure 2—figure supplement 2*), suggesting that cilia on the same cell beat at a similar frequency. Taken together, the presence of local frequency patches rules out global synchronization in our system but still complies with local synchronization across neighboring cells.

We asked next if cilia indeed synchronize, that is, display not only the same CBF, but keep a fixed phase relation. To analyze cilia synchronization in the zebrafish nose, we adopted a measure inspired from neuroscience, the magnitude-squared coherence (*Engel et al., 2001*; *Carter et al., 1973*; *Diaz Verdugo et al., 2019*). This measure, which has also been referred to as cross-correlation in Fourier space, reports the correlation of two signals across frequencies, independent of a phase lag between them. In theory, we should observe high coherence at the CBF only for synchronized cilia with fixed phase relationship, and not for unsynchronized cilia beating at the same frequency accidentally without fixed phase relation (*Figure 2D*).

To measure the degree of coherence between pixels and hence cilia, we calculated the coherence score for one reference pixel with all other pixels in the recording, revealing so-called coherence domains surrounding the reference pixel (*Figure 2E*). To relate coherence to CBF, we recovered the spectral power for the peak frequency of the reference pixel across the entire map (*Figure 2F*) and compared the coherence score to the spectral power for all pixels of the recording (*Figure 2G, H*). We observed that the peak frequency of the reference pixel is dominant in the entire coherence domain (note the similarity in coherence and frequency domains in *Figure 2E–F* also shown in the red area in *Figure 2G–G'* & *Figure 2—figure supplement 3*), confirming that cilia beat at (approximately) the same frequency when synchronized. Occasionally, the frequency domain extended to a region not in the coherence domain (blue highlight in *Figure 2G–G'* & *Figure 2—figure supplement 3*), suggesting that cilia can beat at the same frequency by coincidence. As expected, if two pixels or

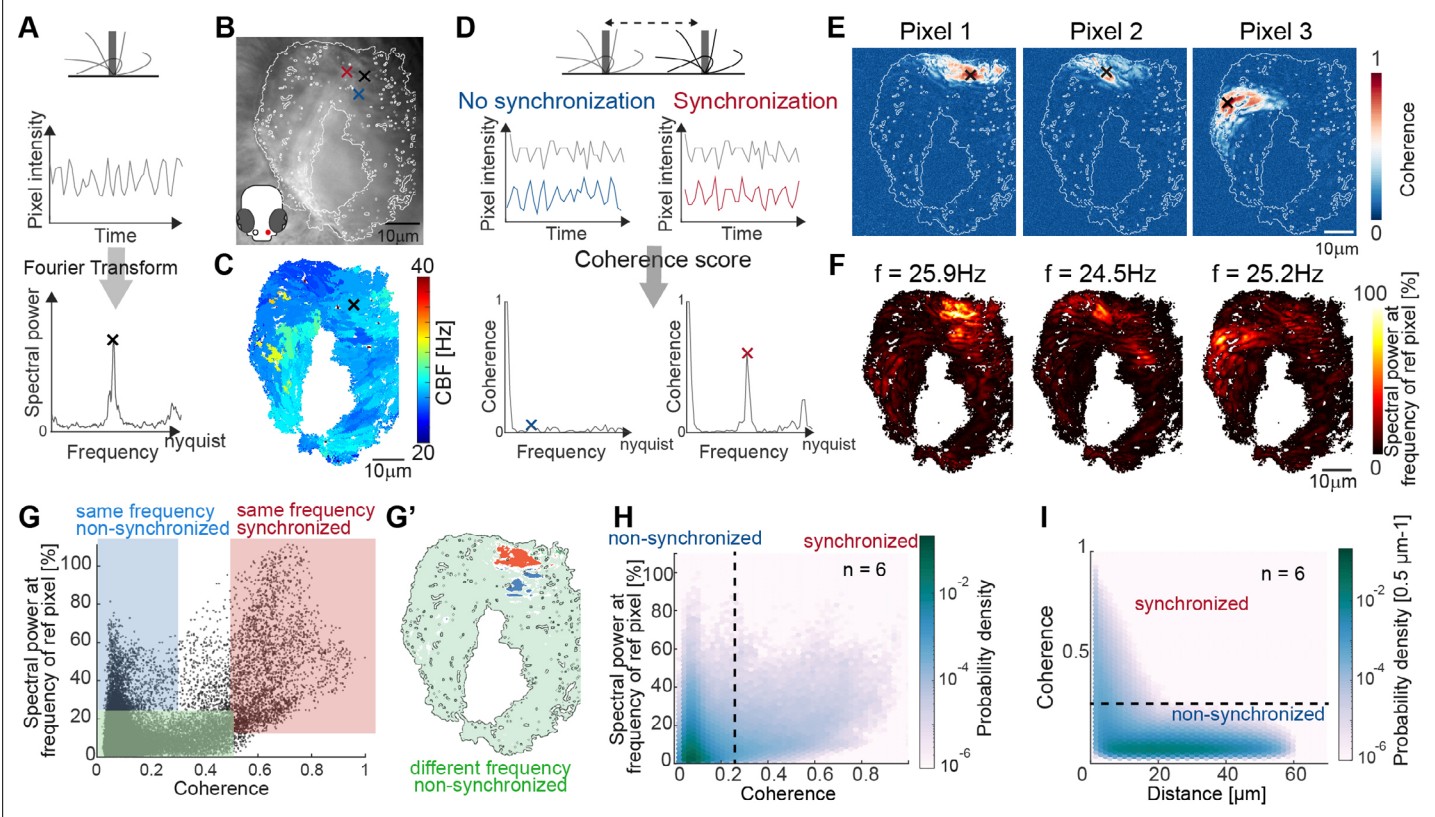

**Figure 2.** Spectral analysis of cilia beating reveals local coherence but global heterogeneity. (**A**) Schematic spectral analysis of a reference pixel. As cilia move through a pixel (black rectangle), the pixel intensity fluctuates. The Fourier transform of pixel intensity time series (top), with peak frequency indicated (bottom). (**B**) Raw image frame of a representative light transmission recording in the left nose of a 4-day-old zebrafish larva overlaid with region representing cilia beating (white line). Example pixels used for panel D are shown with crosses. (**C**) Frequency map of nose pit depicting peak frequency for each pixel. Reference pixel used for panel D is shown with a black cross (**D**) Schematic depicting how the peak coherence measures ciliary synchronization. Note that unsynchronized pixels (blue) have low coherence throughout the frequency spectrum (left), while synchronized pixels (red) have a high coherence at the ciliary beating frequency (right). The location of the color-coded example pixels is shown on panel B (black: reference, blue: not synchronized, red: synchronized). (**E**) Peak coherence for three reference pixels (indicated with black crosses) with all other pixels in a recording. (**F**) Spectral power evaluated at the frequency of the reference pixels ($f$=25.9 Hz; 24.5 Hz; 25.2 Hz) (**G**) Relationship between coherence and spectral power for a representative example (using Pixel 1 from panel E as reference pixel). Three regions of interest are identified: synchronized pixels with high coherence and high spectral power at the frequency of Pixel 1 (red, coherence ≥0.5 and spectral power ≥10%), non-synchronized pixels with high spectral power at the frequency of Pixel 1 but low coherence (blue, coherence ≤0.3 and spectral power ≥25%), and non-synchronized pixels with low spectral power at the frequency of Pixel 1 and low coherence (green, coherence ≤0.5 and spectral power ≤25%). Note that very few pixels show low spectral power but high coherence. (**G'**) Spatial position of the pixels classified in (**G**): Note that synchronized (red) and non-synchronized (blue) pixels do not spatially overlap. Same color scheme in G and G'. (**H**) Analogous to (**G**), but now as average across 6 fish represented as a probability density (using 6 reference pixels per fish). (**I**) The relationship between coherence and pixel distance plotted as probability density for an average of 6 fish. Note that pixels located within 20 μm tend to be more coherent.

The online version of this article includes the following figure supplement(s) for figure 2:

**Figure supplement 1.** Distribution of ciliary beat frequencies.

**Figure supplement 2.** Cilia from individual cells beat at similar frequencies.

**Figure supplement 3.** Systematic analysis of relationships between coherence and spectral power.

**Figure supplement 4.** Cilia beating displays local coherence in the zebrafish brain.

cilia are not oscillating at the same frequency, they are not synchronized (green highlight, *Figure 2G* & *Figure 2—figure supplement 3*).

Since the coherence appeared to be organized in domains (*Figure 2E–G'*), we next calculated the coherence for all signal pixels in the recording and related it to the Euclidian distance separating the two pixels. We saw that, similarly to the reference pixel analysis (*Figure 2E*), the coherence score quickly drops at approximately 20 μm (*Figure 2I*). In comparison, the diameter of a cell is 4.1–4.7 μm.

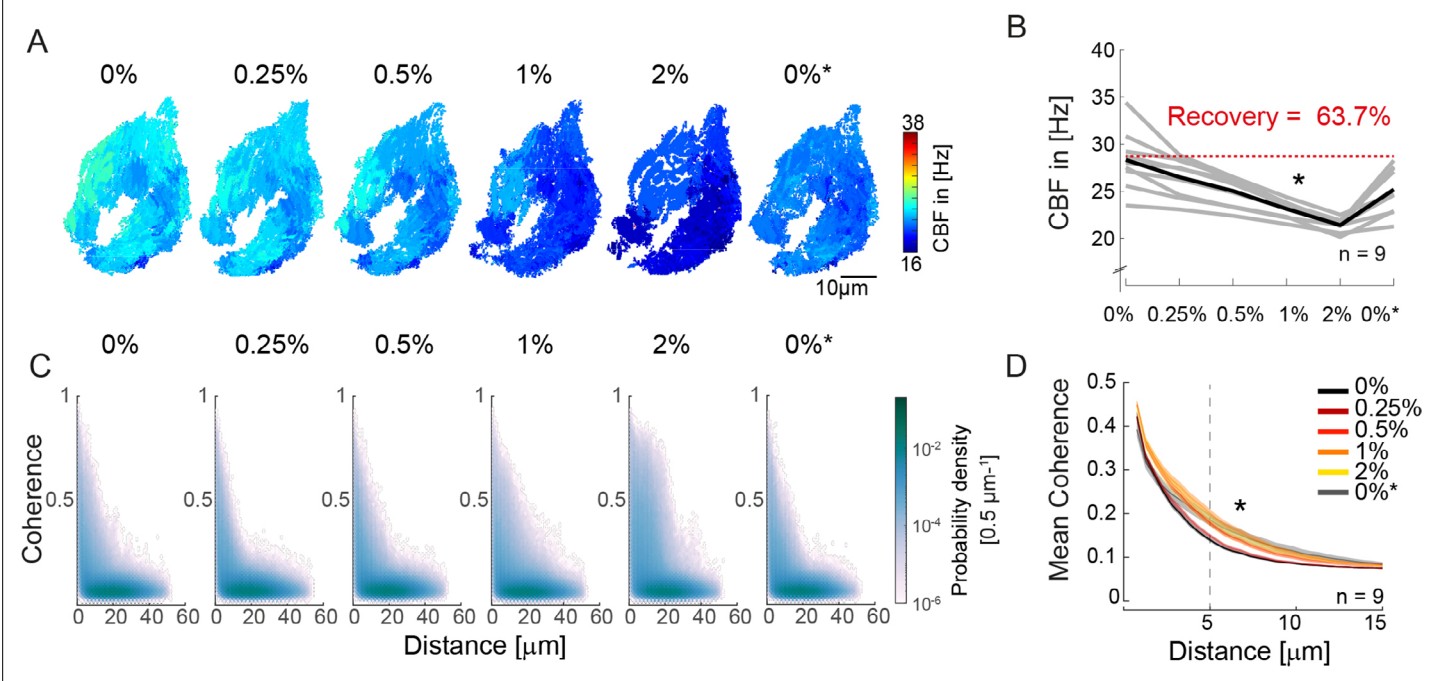

**Figure 3.** An increase in fluid viscosity decreases ciliary beat frequency and extends the spatial range of cilia coherence. (**A–B**) Ciliary beating frequency decreases under increasing viscosity conditions (0–2% methylcellulose) and partially recovers upon re-exposure to 0% methylcellulose (0%*). (**A**) Representative example of ciliary beat frequency (CBF) maps of a 4-day-old zebrafish nose. (**B**) CBF for n=9 (gray) and average in black. A repeated measures ANOVA (*) indicates a significant effect of viscosity conditions on CBF (p = 0.003; n = 9). (**C–D**) Ciliary coherence extends with increased fluid viscosity. (**C**) A representative example of pairwise coherence versus distance for different viscosity conditions (Coherence bin width = 0.04/bin; distance bin width = 0.5 μm/bin). (**D**) Mean coherence across distance bins (width = 0.5 μm) for different viscosity conditions shows that coherence domains expand for conditions of increased viscosity. Mean curves and standard error of the mean are plotted (n=9). ANOVA-N indicates a significant effect of viscosity conditions on mean coherence across distances (p=3·10⁻⁵). CBF = Ciliary Beat Frequency.

The online version of this article includes the following figure supplement(s) for figure 3:

**Figure supplement 1.** Ciliary beating in different viscosity conditions.

We next performed similar analyses in another multiciliated tissue located in the zebrafish adult brain, the tela choroidea (**D'Gama et al., 2021**, **Jeong et al., 2022**). Similarly to the nose pit, we observed high heterogeneity of CBF restricting synchronization locally (**Figure 2—figure supplement 4**). Notably, we identified that the coherence score quickly drops at approximately 30 μm (**Figure 2— figure supplement 4**).

Altogether, we propose the coherence score and the coherence versus distance as methods to quantify synchronization within a cilia carpet. Application of these methods to the multiciliated epithelia of the zebrafish nose pit and adult brain indicates that cilia synchronize locally rather than globally.

## Increasing viscosity reduces ciliary beating frequency and extends ciliary coherence

Since cilia synchronization appears to be local in nature, we asked whether changes in cilia beat dynamics can influence the spatial extent of their synchronization. To test this, we exposed the zebrafish nose to surrounding mediums of increasing viscosity (1,875 cP; 3.75 cP; 7.5 cP; 15 cP). We chose to apply methylcellulose based on its reversible and non-invasive properties, and prior work suggesting that viscosity change can affect synchronization between cilia (**Gheber et al., 1998**; **Machemer, 1972**). By increasing methylcellulose concentration in a stepwise manner, we found that the CBF decreases upon increases in viscosity of the surrounding medium (**Figure 3A–B** & **Figure 3— figure supplement 1B**), similarly to previous reports (**Brokaw, 1966**; **Katoh et al., 2018**; **Machemer, 1972**).

Next, to probe the impact of viscosity on synchronization, we calculated the coherence versus distance for all pixels in the recordings (*Figure 3C*). To visualize changes in the distribution, we plotted the difference from the baseline for the different conditions (*Figure 3—figure supplement 1A*) and observed that the number of coherent pixel pairs increased in the 5–20 µm distance range when viscosity increased. To quantify this effect, we compared the mean coherence (*Figure 3D*) and the fraction of coherent pixels using a threshold of coherence >0.25 (*Figure 3—figure supplement 1B*) across distance bins (width = 0.5 µm). We observed a significant effect of high viscosity conditions for both measures, suggesting that increasing viscosity enhances synchronization between pixels with distance in the 5–20 µm range.

Increasing the viscosity of the surrounding medium and thus the hydrodynamic load acting on beating cilia was previously shown to change the waveform of cilia beating (*Brokaw, 1966*; *Katoh et al., 2018*; *Klindt et al., 2016*), a phenomenon termed waveform compliance (*Goldstein et al., 2016*). While this could in principle explain the improved synchronization observed in the zebrafish nose (*Figure 3D*, *Figure 3—figure supplement 1B*), changes in cilia waveform, if present, were too small to detect. Specifically, we performed light-sheet microscopy on multiciliated cells sparsely expressing GFP in their cilia upon increasing viscosity conditions. The cells were imaged from a transverse angle to allow manual reconstruction of the cilia beating waveform. By applying our FFT-based analysis on the light-sheet recordings, we observed a decrease in CBF, matching the light transmission experiment (*Figure 3—figure supplement 1C, D*). Similarly, kymographs revealed changes in the frequency and kinetics of ciliary beating (*Figure 3—figure supplement 1E–E'*). Manual tracking of cilia waveforms did not reveal obvious alterations in the ciliary beat amplitude, although our approach may not be sensitive enough to observe slight variations in waveform (*Figure 3—figure supplement 1F*). We argue that the combined use of coherence and frequency analyses is better suited for investigating changes in the collective cilia dynamics than the use of kymographs and manual tracing. Using this combined approach, our results indicate that an increase in viscosity both changes cilia beating properties and improves short-range synchronization.

## Persistent tissue-scale patterns of metachronal coordination differ for left and right noses

Motile cilia can coordinate their beat into so-called metachronal waves, which refer to the sequential beating of neighboring cilia (*Gray, 1922*; *Brumley et al., 2015*; *Wan et al., 2020*; *Ovadyahu and Priel, 1989*; *Nawroth et al., 2017*). Given the heterogeneous beating frequency and lack of global synchronization that we observed, we asked whether metachronal waves emerge in our system.

Metachronal waves have been commonly detected using kymographs (*Brumley et al., 2015*; *Wan et al., 2020*). Using such an approach on our light transmission recordings (*Figure 4A–A'*), we observed that metachronal-like activity is present in the zebrafish nose, but is not stable over time, as shown by analyzing the same recording during two different time windows (*Figure 4A'*). This suggests that these waves are highly dynamic in space and time and studying them by means of kymographs is time-intensive and prone to subjective interpretations. In agreement with our findings, simulated metachronal waves may display defects (*Elgeti and Gompper, 2013*), but maintain a stable direction over time (*Elgeti and Gompper, 2013*; *Solovev and Friedrich, 2022b*). Hence, we developed an automated detection method to quantify and visualize the properties of metachronal waves, including direction and wavelength, based on the phase angle of our FFT-based CBF detection method (*Figure 4B*).

To obtain robust phase angles across the ciliated epithelium, we first segmented the frequency map into distinct patches, so that neighboring pixels with similar frequency (CBF bin = 0.54 Hz) were assigned to the same patch. We then extracted the phase angle from the Fourier spectrum for the dominant frequency of that given patch. We decided to extract phases from segmented patches because (*i*) phases extracted for different frequencies are not comparable and lead to noisy signals precluding further analysis, and (*ii*) for time series of finite length, phases of Fourier modes at nearby frequencies are correlated. This analysis provides a unique phase angle for each pixel of the recording (*Figure 4B* right and *Videos 1 and 2*). Next, to translate phase information to wave direction and wavelength quantifications, we extracted image gradient vectors from each patch separately (*Figure 4C*). This allowed us to quantify and plot both wave direction and wavelength, by calculating the mean gradient vector direction and the mean gradient vector length, respectively (*Figure 4D and E*). Using this

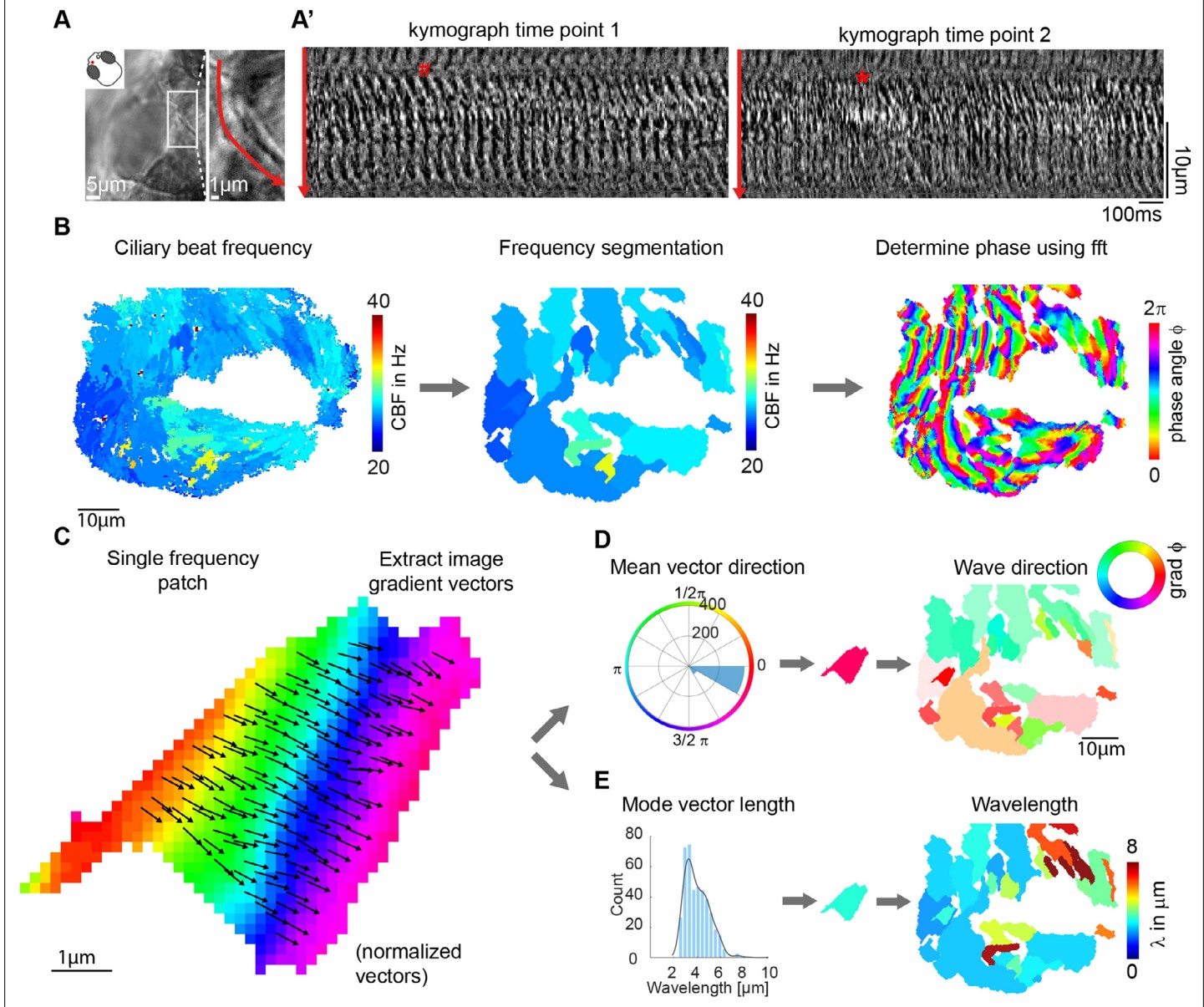

**Figure 4.** Wave directions and wavelengths of local metachronal coordination. (**A-A'**) Metachronal coordination observed using a conventional kymograph-based analysis. (**A**) A kymograph was drawn (red line in inset, representing transverse cilia beating) on a light transmission recording of a zebrafish nose at 4dpf. (**A'**) Kymographs of cilia beating in the same location at different time points. Note the orderly pattern in the left panel (#) versus the disorderly pattern in the right panel (*). (**B–D**) Pipeline to measure metachronal coordination based on a phase angle method. (**B**) Neighboring pixels with similar frequency (beat frequency map, left) are segmented into patches (center). Phase angles are determined from Fourier transforms evaluated at the prominent frequency of each segmented frequency patch (right). (**C**) Analysis proceeds for each patch by extracting an image gradient, as shown by the arrows. (**D–E**) The mean direction of the gradient vector characterizes wave direction – with transparency representing the inverse circular standard deviation (D, right) while its length determines the wavelength (E, left). Scale bars, 10 µm. See also **Videos 1–3**.

The online version of this article includes the following figure supplement(s) for figure 4:

**Figure supplement 1.** Metachronal wave directions and lengths.

**Figure supplement 2.** Cilia beating displays local metachronal coordination in the ependymal layer of the zebrafish brain.

approach, we found that the wave directions form regular patterns (**Figure 4D**), which are relatively stable over time (**Figure 4—figure supplement 1A** and **Video 3**), across depth (**Figure 4—figure supplement 1B**), and not particularly affected by increasing viscosity (**Figure 4—figure supplement 1C**). In contrast, we observed a large variability of wavelength in space and time (**Figure 4—figure**

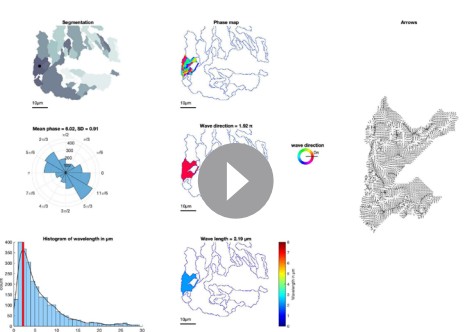

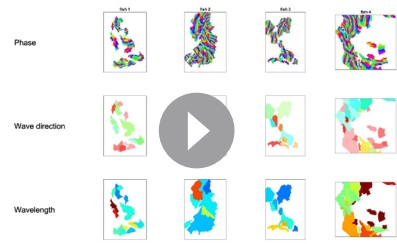

**Video 1.** Measurement of metachronal wave properties in segmented frequency patch.
https://elifesciences.org/articles/77701/figures#video1

**Video 3.** Metachronal wave direction is stable over time (total of 10 min) as shown for four examples. Every frame of the video corresponds to the output of a Fourier Transform calculated over a 30 s timebin.
https://elifesciences.org/articles/77701/figures#video3

supplement 1A–C). Similar results were obtained for the tela choroidae in the brain (*Figure 4—figure supplement 2*). The patterns were, however, more diverse, probably due to a lower density of cilia and motile ciliated cells (*D'Gama et al., 2021*). In summary, our results indicate that metachronal waves are present in the zebrafish nose and adult brain and that their wave direction remains stable.

We next asked whether the observed pattern of metachronal wave direction is a consistent feature across different individuals. To compare multiple noses, we first aligned multiple recordings of left noses and found that the pattern of wave directions is highly comparable across noses, while wavelengths vary (*Figure 5A* & *Figure 5—figure supplement 1A-B*). Next, to compare left noses with right noses, we mirrored and aligned right noses so that they had the same anterior-posterior and mediolateral orientation as the left noses. Surprisingly, we observed that the wave directions between left and right are consistently different for the lateral side of the noses (*Figure 5B* & *Figure 5—figure supplement 1A-B*). To relate the direction of the metachronal waves to cilia beating, we measured the cilia beat direction. The cilia beat direction is set developmentally (*Guirao et al., 2010*; *Gsell et al., 2020*; *Vladar et al., 2012*; *Wallingford and Mitchell, 2011*; *Mitchell et al., 2009*) and can be inferred from the location of the cilia basal foot, which points toward the cilia beat direction (*Clare et al., 2014*; *Ramirez-San Juan et al., 2020*). To discern the cilia beat direction for each cell, we stained the 4dpf zebrafish nose for basal feet (gamma-tubulin) as well as the cilium (glutamylated tubulin) (*Figure 5C* and *Figure 5—figure supplement 2*). We found that the ciliary beat direction is consistent across fish (*Figure 5D*) and is mirrored for the left and right noses (*Figure 5D& E*). In contrast, we observed a stark difference in the angle of the metachronal wave, with an angle of 163.2° relative to the anterior-posterior axis for left noses (*Figure 5F*) and 100.2° for the mirrored right noses (*Figure 5G*). We did not observe any significant difference in any other measures between the left and the right noses, including fields of fluid flow (*Figure 5—figure supplement 3*, *Figure 5—figure supplement 4*). Ciliary beating of multiciliated cells have been shown to have a certain rotational component, which introduces a defined chirality of the cilia beat (*Machemer, 1972*). Moreover, in modeling studies, this chirality was shown to set a direction of emergent metachronal waves different from the cilia beat direction (*Elgeti and Gompper, 2013*; *Solovev and Friedrich, 2022b*). The chiral cilia beat patterns explains why the direction of metachronal waves can be rotated relative to the main cilia beat direction. However, it cannot explain why we observe different angles for the left and right noses.

In conclusion, by designing novel ways to quantify cilia synchronization and the spatial organization of cilia within a tissue in dense carpets,

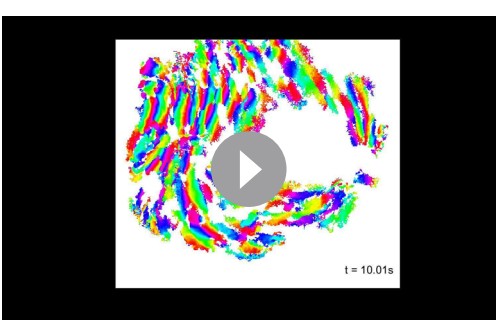

**Video 2.** Measurement of metachronal wave in a 30 s long recording. Each frame of the movie represents the analysis of 20s-long sliding windows. The timer indicates the center of the sliding window.
https://elifesciences.org/articles/77701/figures#video2

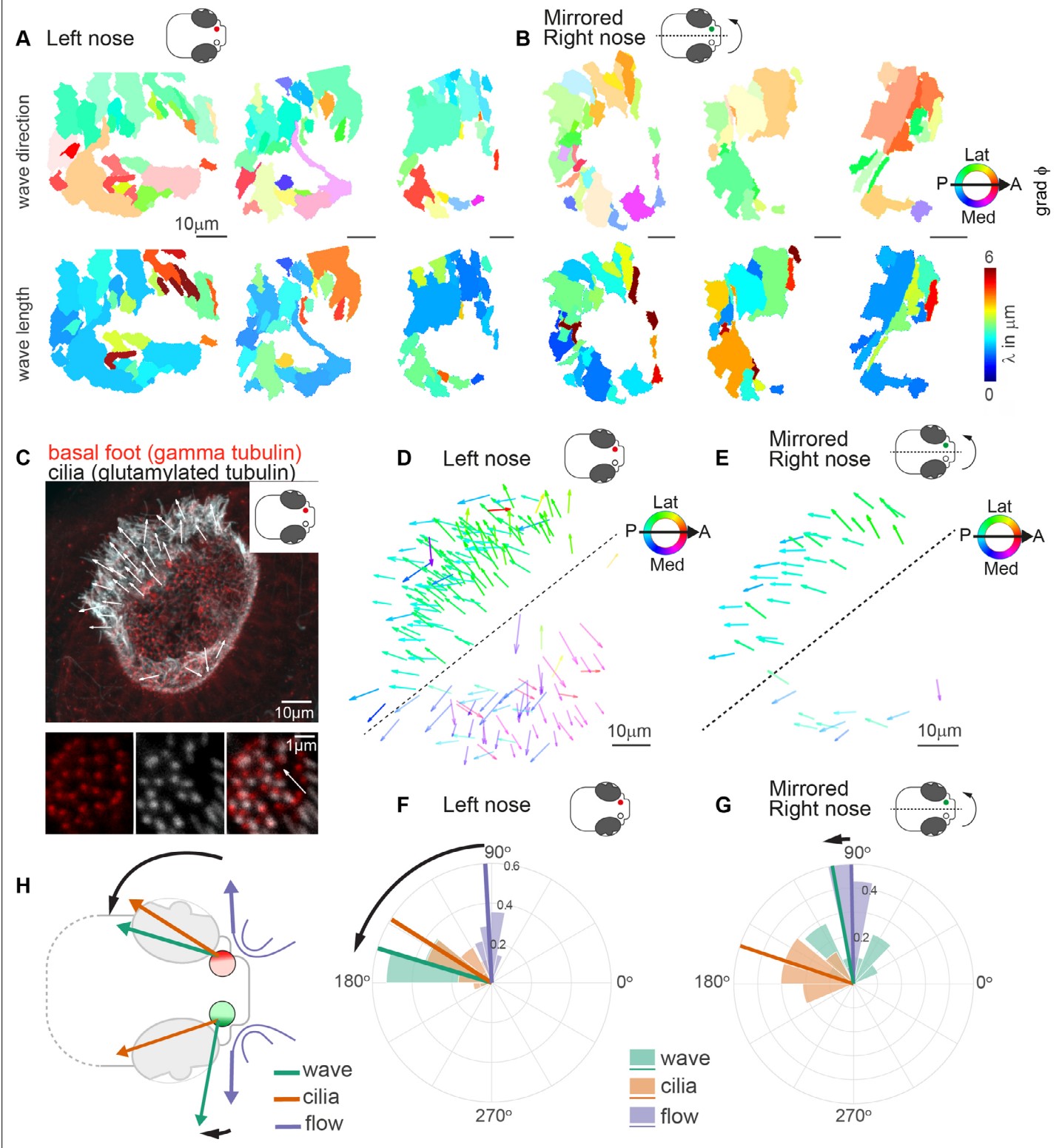

**Figure 5.** Metachronal waves are chiral. (**A–B**) Wave direction (top) and wavelength (bottom) for three left (**A**, red) and three mirrored right (**B**, green) noses show asymmetry in the wave direction between the left and right noses. Transparency reflects the inverse circular standard deviation. (**C**) Immunohistochemistry on a left nose stained for gamma-tubulin (basal body marker, red) and glutamylated tubulin (cilia marker, white). Zoom-in obtained at higher magnification displays how gamma-tubulin and glutamylated-tubulin stains are offset, allowing to determine cilia foot orientation and thus ciliary beat direction. (**D–E**) Overlay of all ciliary beat directions in the left (**D**; n=10) and mirrored right (**E**; n=3) noses. Individual arrows refer

*Figure 5 continued on next page*

*Figure 5 continued*

to the polarity of individual cells across fish. Direction is color-coded. Note a clear distinction in polarity between the latero-posterior and medial part of the nose indicated by a dashed line. (**F–G**) Quantification of ciliary beat directions, metachronal wave (left, n=16; right, n=18) and overall fluid flow directions for left (**F**; n=14) and mirrored right (**G**; n=14) noses. Plotted are the mean directions per fish for the latero-posterior part of the noses (above dashed line in D and E). Note that a direct comparison of ciliary beating direction and wave direction in the same experiments was not possible due to different positioning of the zebrafish for both experiments. (**H**) Schematic of the ciliary beating, metachronal wave and fluid flow directions in the left versus the right noses. Note the offset between the fluid flow (blue) and metachronal waves directions (green) for the left and right noses. Scale bars 10 μm.

The online version of this article includes the following figure supplement(s) for figure 5:

**Figure supplement 1.** Difference in wave direction between left and right noses.

**Figure supplement 2.** Cilia orientation is mirrored in the left and right noses.

**Figure supplement 3.** No anatomical differences between left and right noses.

**Figure supplement 4.** Sequential measurement of ciliary beating and fluid flow direction.

we revealed the presence of metachronal waves with stable wave directions that are different for the left and right noses.

## Metachronal coordination enhances fluid pumping and reduces steric hindrance but does not affect direction of fluid flow

To investigate the role of metachronal coordination in cilia carpets on their physiological function of fluid pumping, we resorted to a computational model. Specifically, we used a recently established multi-scale model of a regular array of cilia (*Solovev and Friedrich, 2022b*; *Solovev and Friedrich, 2022a*), which uses detailed hydrodynamic computations and an experimentally measured three-dimensional beat pattern from *Paramecium* (*Machemer, 1972*). This model can probe the effect of any wave direction and wavelength of metachronal coordination on fluid pumping and steric interactions, i.e., cilia colliding with each other. Thereby it allows us to disentangle the roles of hydrodynamic interactions and the importance of avoiding steric collision between neighboring cilia in a systematic way. The computational model enables us to investigate metachronal waves with different directions and wavelengths in a cilia carpet as shown in *Figure 6A*, where each dot in the hexagonal plot represents a different wave solution with different wave length $\lambda$ and direction angle θ represented by a wave vector $(k_x, k_y) = 2\pi/\lambda$ ( - sin θ, cos θ). Using this model, we first investigated how different traveling waves affect the pumping rate per cilium and the mean direction of fluid flow. We observed that the pumping rate is close to its minimum when all cilia beat with the same phase, that is, for in-phase cilia beating with $(k_x, k_y) = (0,0)$ (*Figure 6B*). In contrast, if cilia beat in the form of a metachronal traveling wave with finite wavelength $\lambda$, the pumping rate increases significantly. We observed a stronger increase in pumping rate for symplectic or antiplectic metachronal coordination (for which the wave vector is parallel to the direction of the cilia effective stroke, $\theta = 0^0$ or $180^0$), as compared to dexioplectic or laeoplectic metachronism (for which the wave vector is perpendicular to the direction of the effective stroke, θ = ±90$^0$). The observed change in pumping rate is a result of hydrodynamic interactions between cilia: if cilia move in opposite directions for an antiplectic wave, destructive interference of flow fields implies that the hydrodynamic load increases for each cilium. Thus, the beating cilia perform more work on the fluid per beat cycle and consequently pump the fluid faster. Hydrodynamic interactions also explain why the pumping rate is higher for symplectic and antiplectic waves as compared to laeoplectic and dexioplectic waves because hydrodynamic interactions are stronger in the direction of an applied force as compared to the direction perpendicular to the force. As a subtle point, an increase in hydrodynamic load also slows down the collective frequency of cilia beating (*Solovev and Friedrich, 2022b*), but this secondary effect is not strong enough to revert the increase of pumping rate due to enhanced hydrodynamic load. For numerical reasons, our computational model allows only a dilute cilia density of 0.0036 μm$^{-2}$ for which cilia do not collide. For higher cilia densities, we expect an even more pronounced change in pumping rate (*Elgeti and Gompper, 2013*; *Osterman and Vilfan, 2011*).

Next, we investigated the impact of different metachronal wave solutions on the direction of cilia-generated fluid flow. As shown in *Figure 6C*, we observed that the flow direction is virtually independent of the direction and wavelength of metachronal waves with a variability of less than ±5$^0$ for the

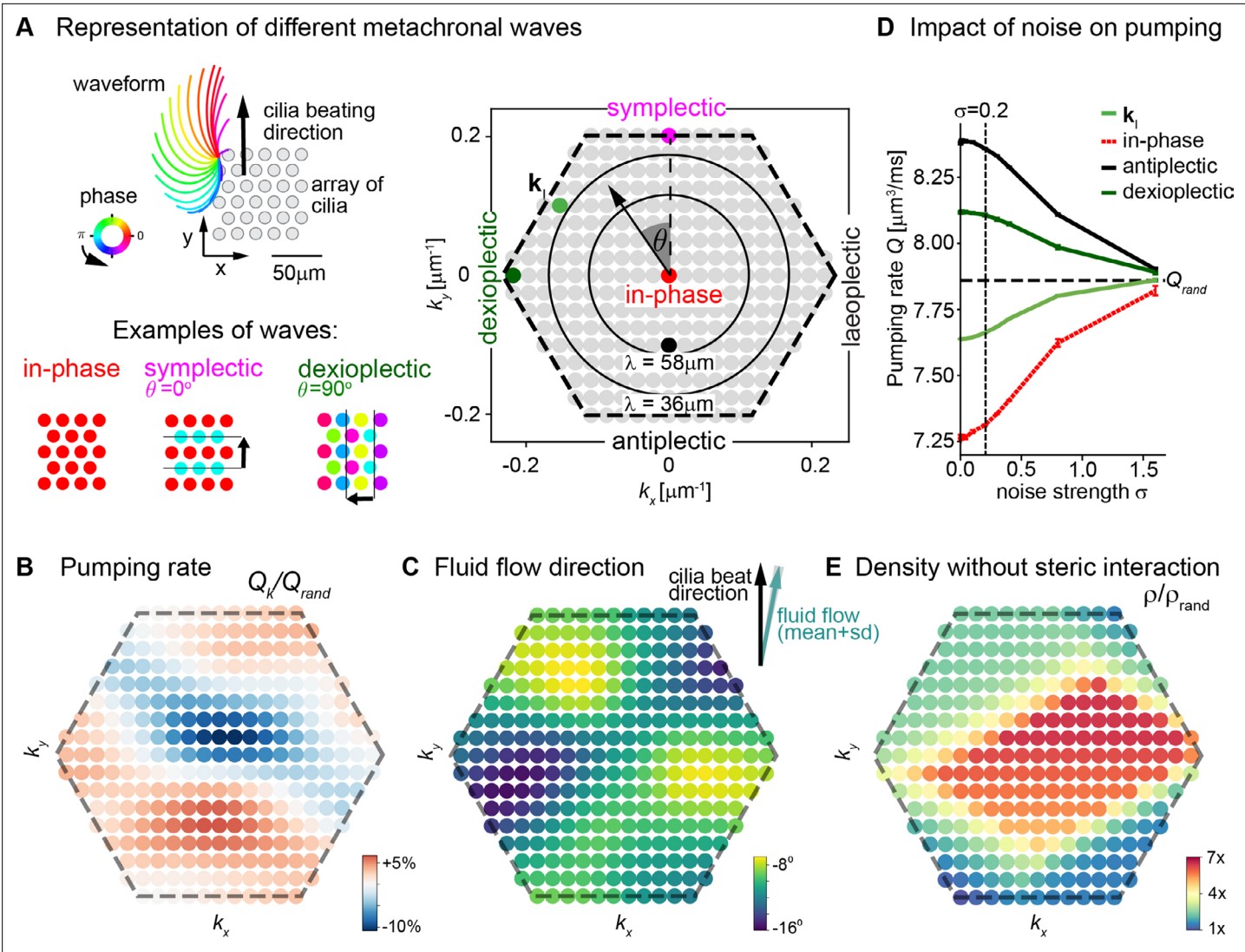

**Figure 6.** Metachronal coordination enhances fluid pumping and reduces steric interactions, but does not affect fluid flow direction. (**A**) Possible traveling wave solutions in a computational model of a cilia carpet. Left: Cilia are arranged on a triangular lattice (gray dots), with three-dimensional cilia beat pattern from *Paramecium* (not to scale, cilia length 10 μm). Three example wave solutions are highlighted: in-phase beating, symplectic wave, dexioplectic wave. Wave fronts are indicated by black lines and the wave direction by an arrow, while the color code of cilia base points represents cilia phase. Right: Visualization of the set of all possible wave solutions as function of wave vector $\mathbf{k}=(k_x, k_y)$, where the distance from the origin encodes the wavelength of the wave as $\lambda = 2\pi / |\mathbf{k}|$ and the directional angle $\theta$ encodes the direction of the wave relative to the direction of the effective stroke of the cilia beat. Example waves from left panel are highlighted as colored dots. (**B**) Pumping rate $Q$ per cilium computed for different wave solutions, with each wave represented by a color-coded dot as in hexagon plot of panel A (normalized by pumping rate $Q_{rand} \approx 7.87$ μm³/ms for cilia beating with random phase relationship). (**C**) Direction of cilia-generated flow (averaged over one beat cycle) for different wave solutions relative to the direction of the effective stroke of the cilia beat; note that the range of the color bar spans only $10^0$. Mean ± standard deviation is shown in the upper right corner. (**D**) Pumping rate per cilium for four selected wave solutions as indicated in hexagon plot of panel (**A**) as function of a noise strength σ (see text for details). Dashed line indicates mean pumping rate for cilia beating with random phase relationship ($Q_{rand}$). (**E**) Critical density $\rho$ of a cilia carpet below which steric interactions between cilia arise for different wave solutions; density $\rho$ is normalized relative to a critical density for cilia beating with random phase relationship, $\rho_{rand} = 0.015$ μm⁻².

The online version of this article includes the following figure supplement(s) for figure 6:

**Figure supplement 1.** Visualization of increasing noise strength on synthetic metachronal waves.

chosen cilia density. This theory prediction matches our experimental observation. Indeed, we did not find any difference in the overall fluid flow direction in left and right noses with their different metachronal directions (*Figure 5F&G* and *Figure 5—figure supplement 4*). In conclusion, our results show that metachronal coordination is beneficial for fluid pumping, but does not affect the direction of fluid flow.

We then probed how deviating from global synchronization impacts fluid pumping. We previously demonstrated theoretically that above a characteristic level of cilia phase noise, global synchronization is lost but local synchronized domains persist (*Solovev and Friedrich, 2022a*). For efficient numerical computations, it is sufficient to consider a smaller carpet that exhibits a single coordinated metachronal wave perturbed by cilia phase noise, instead of several domains with local synchronization each. This is valid because long-range hydrodynamic interactions near a boundary wall such as an epithelial layer decay as inverse cubed distance ($\sim d^{-3}$). Therefore, hydrodynamic interactions between distant cilia should have little impact on the rate of fluid pumping, and in fact were previously shown to be dispensable for our computational model (*Solovev and Friedrich, 2022b*). Hence, we expect that in a cilia carpet with noise-perturbed global synchronization, the main determinant affecting fluid pumping is phase noise between nearby cilia. Correspondingly, we investigated how adding increasing phase noise to a prescribed metachronal wave affects fluid pumping for four possible scenarios: (*i*) the dexioplectic wave with wave vector $\mathbf{k}_I$ that emerges spontaneously in the computational model (*Solovev and Friedrich, 2022b*), (*ii*) a perfect dexioplectic wave with θ = 90°, (*iii*) the antiplectic wave that maximizes fluid pumping, and (*iv*) in-phase cilia beating.

We observed that, for the dexioplectic wave $\mathbf{k}_I$, the rate of fluid pumping approximately equals that for cilia beating with a random phase relationship, while the pumping rate is substantially higher for the most efficient antiplectic wave and a dexioplectic wave with θ = 90° (*Figure 6D*). Generally, if random perturbations of cilia phase remain small (standard deviation σ<0.2, *Figure 6—figure supplement 1*), the pumping rate is only moderately changed. For stronger phase noise (σ>1, *Figure 6—figure supplement 1*), the pumping rate attains a constant value, irrespective of the underlying wave solution, as cilia essentially beat with random phase relationship. Taken together, our results show that cilia carpets exhibiting metachronal coordination can tolerate some level of noise of cilia beating for their physiological function of fluid pumping.

Besides improving fluid flow, metachronal waves may also prevent steric collisions between neighboring cilia, which could slow down their dynamics. To probe the relevance of steric interactions, we computed for each wave solution a critical density $\rho(\mathbf{k})$ at which cilia do not collide (*Figure 6E*). We observed that this critical density is substantially higher than a corresponding density $\rho_{rand}$ defined for cilia beating with random phase relationship. In fact, this critical density is minimal for in-phase cilia beating. However, for in-phase beating, the predicted pumping rate is also minimal. In contrast, symplectic and dexioplectic metachronal coordination allow for both an elevated pumping rate and denser cilia packing without steric collisions. The product of the pumping rate per cilium and cilia density yield the total pumping rate of a cilia carpet. We speculate that metachronal coordination may represent an evolutionary solution to the trade-off choice of both realizing a high pumping rate per cilium (which favors metachronal coordination with short wavelength, see *Figure 6B*) and a high cilia density (which favors in-phase synchronization, see *Figure 6E*). Our argument refines a previous hypothesis by Machemer, who proposed that dexioplectic waves reduce steric interactions as compared to symplectic or antiplectic waves *Machemer, 1972*.

Our computational model shows that local synchronization is a prerequisite for efficient cilia beating in dense cilia carpets by avoiding steric collisions between cilia. We propose that local synchronization allows to already realize almost the maximal possible rate of fluid pumping, while reducing steric interactions effectively.

## Discussion

By using novel tools to quantify ciliary synchronization in vivo and a computational model, we put forward the notion of local but not global synchronization as the expected form of cilia coordination in the presence of noise and perturbations. Even with only local synchronization of cilia, tissue-scale metachronal coordination with distinct pattern of wave directions is possible in cilia carpets. Indeed, we observed in a zebrafish model system tissue-scale patterns of metachronal waves, which were consistent across individual animals, but different for left and right noses. This difference in wave

directions is not a mere consequence of different patterns of cilia alignment, but instead points at an important role of the chirality of the cilia beat and 3D architecture of the underlying tissue. On the basis of computational results, we propose that local synchronization is sufficient to pump fluids at an almost maximal rate.

In this study, we chose to study multiciliated cells in the developing zebrafish nose. We identified that although these cells bear a reduced number of cilia as compared to mammalian tissues or *Xenopus* embryonic skin, they retain a ciliary density similar to these other multiciliated epithelia due to their smaller apical area (*Goldstein et al., 2016*). In principle, the apical size of cells should not matter for the physics of cilia synchronization. As a small caveat, it is possible that a ciliated epithelium made of smaller cells may be more heterogeneous due to higher divergence in cilia properties across cells. Yet, since the density of cilia is a key property for synchronization of motile cilia (*Brumley et al., 2014*; *Pellicciotta et al., 2020*), we argue that the zebrafish nose is an appropriate model for studying coupling mechanisms within and across cell boundaries in a naturally dense cilia carpet, which is highly accessible for imaging experiments. We have also performed experiments in another ciliated tissue, the tela choroidea of the adult zebrafish brain (*D'Gama et al., 2021*, *Jeong et al., 2022*), to further strengthen the applicability of our analysis tools.

Using Fourier-based analyses, we observed that motile cilia in the zebrafish nose and brain beat at slightly different frequencies, both within the organ and across fish. The beating frequency may depend on the number of cilia on the cell (*Pellicciotta et al., 2020*), the age of the cell or animal (*Olstad et al., 2019*), but likely not the length of the cilium (*Bottier et al., 2019*; *Pintado et al., 2017*). Beat frequency may also relate to the level of synchronization across cilia. In fact, it was shown that upon synchronization, the emergent frequencies of two oscillators converge to the same value as long as the intrinsic frequencies are similar enough (*Brumley et al., 2014*; *Pellicciotta et al., 2020*; *Quaranta et al., 2015*; *Pikovsky et al., 2003*). Moreover, cilia should beat faster (or slower) when they are synchronized in-phase (or in anti-phase) because the hydrodynamic friction acting on an individual cilium decreases (or increases) (*Elgeti and Gompper, 2013*; *Pellicciotta et al., 2020*; *Solovev and Friedrich, 2022b*; *Meng et al., 2021*). Of note, the cilia beat frequencies reported here do not represent the intrinsic beat frequencies of individual cilia, but instead the emergent frequency when a cilium is interacting with its neighbors. Nevertheless, in our system, we observed that beat frequencies are widely distributed, suggesting that there is no synchronization on a global, tissue scale. This also agrees with experimental data in airway human cells (*Feriani et al., 2017*) and modeling work showing that frequency dispersity and active noise in cilia beating limit synchronization to the local scale (*Guirao and Joanny, 2007*; *Solovev and Friedrich, 2022a*). To probe how global versus local synchronization affects the transport of fluid, we turned to a computational approach. We highlight how local synchronization is a prerequisite to reduce steric hindrance in dense cilia carpets. Moreover, the pumping rate of a cilia carpet is robust against moderate levels of phase noise between neighboring cilia, while it should be independent of hydrodynamic interactions with distant cilia, suggesting that locally synchronized cilia can achieve fluid pumping at an almost maximal possible rate. Our results thus complement a previous study suggesting a favorable effect of slight disorder in the cilia polarity for effective fluid transport in the lung (*Ramirez-San Juan et al., 2020*), by highlighting benefits of local order.

Previous methods to investigate coordinated ciliary movement include spatial correlation approaches (*Gheber and Priel, 1989*; *Wan et al., 2020*), measures of phase similarity (*Oltean et al., 2018*), or spatial correlation functions (*Feriani et al., 2017*; *Brumley et al., 2015*). However, these approaches are either not automated, or complicated to interpret. In this study, we applied another measure for synchronization commonly used in neuroscience, the magnitude-squared coherence (*Engel et al., 2001*; *Carter et al., 1973*). This measure proved to be very robust to detect synchronization across pixels. Synchronized oscillators should beat at the same frequency. Indeed, we observed that our coherence score only reported pixel pairs oscillating at the same frequency. Yet, having the same frequency is only a necessary but not a sufficient condition. Indeed, we noticed that not all pixel pairs oscillating at the same frequency were coherent, in particularly when pixels were far apart and only 'accidentally' shared the same frequency. Using this measure, we observe that coherence rapidly decays at a pixel distance of approximately 20 µm, which is larger than the diameter of a cell (4.1–4.7 µm), and longer than the length of a single cilium of approximately 9 µm in the nose. We thus have an indirect, but not yet direct proof that coherence domains cross cell boundaries. It remains to

be understood what sets the size of the observed coherence domains. Generally, a stronger coupling between cilia promotes larger domains, while active noise of the cilia beat, dispersity of intrinsic cilia beat frequency, and imperfect cilia alignment should cause smaller domains (*Solovev and Friedrich, 2022b*; *Solovev and Friedrich, 2022a*; *Guirao and Joanny, 2007*).

Our results revealed that coherence domains slightly increase in size when the viscosity of the surrounding medium is increased. This complements previous findings by Machemer that changing viscosity can change the direction of metachronal waves in cilia carpets in unicellular *Paramecium* (*Machemer, 1972*). We suppose that changing the viscosity of the surrounding medium slightly changes the cilia beat pattern and therefore the synchronization coupling between nearby cilia. Indeed, previous literature showed that shape and speed of the cilia beat changes upon increased viscosities or hydrodynamic load (*Brokaw, 1966*; *Klindt et al., 2016*).

Traveling waves of ciliary phase, in the form of so-called metachronal waves, have been previously observed in ciliated system, particularly in ciliates (*Wan et al., 2020*; *Brumley et al., 2015*). They are often highlighted as a feature of synchronized cilia (*Bustamante-Marin and Ostrowski, 2017*; *Ovadyahu and Priel, 1989*), which promote efficient fluid transport (*Elgeti and Gompper, 2013*). Using a novel analysis method based on the phase of the Fourier transform of time-lapse recordings, we revealed coherent patterns of metachronal wave directions in nose pits, which were consistent across individual fish. Intriguingly, wave patterns were different for the left and right noses with an offset of approximately 65° in relation to the anterior-posterior axis of the animal. Of note, due to a slighty different mounting angle of the samples for measuring ciliary beating and cilia polarity, we are not able to directly measure the offset between ciliary beating and metachronal wave direction and hence normalized these values to the anterior-posterior axis of the larvae.

As we did not observe any other differences between the left and right noses, we speculate that chiral cilia beat patterns may be slightly different for the left and right noses, respectively, since different cilia beat patterns will give rise to different dominant metachronal waves (*Meng et al., 2021*), as observed for different metazoan species (*Knight-Jones, 1954*). Usually, only one type of metachronism is observed in a single species, and even in a single systematic group. The zebrafish model system established here thus opens the unique opportunity to study two different fundamental types of metachronal coordination in a single species.

Using hydrodynamic computations, we showed that metachronal coordination hardly changes the direction of cilia-generated fluid flow. Hence, it is unlikely that the different metachronal wave patterns observed between the left and right zebrafish noses and in different species in general affect the direction of fluid flow and thus their physiological function. In line with this, we have not observed differences in the direction of the overall flow generated by the left or right pit. However, metachronal coordination enhances the rate of fluid pumping and at the same time reduces steric interactions at higher cilia density. We speculate that the observed metachronal waves may provide an optimal trade-off between a high pumping rate per cilium and a high cilia density within a cilia carpet.

In conclusion, we describe here an accessible in vivo model system and novel analytical methods to quantify the properties of the cilia beat as well as emergent synchronization in ciliated epithelium in an unbiased manner. Using these tools, we show that local synchronization is sufficient for coherent patterns of metachronal coordination, which expectedly allows to realize almost the maximal rate of fluid pumping.

## Materials and methods

**Key resources table**

| Reagent type (species) or resource | Designation | Source or reference | Identifiers | Additional information |
|---|---|---|---|---|
| Genetic reagent (zebrafish) | *Et(hspGGFF19B:Gal4)Tg(UAS:gfp)* | *Reiten et al., 2017*; *Asakawa et al., 2008* | ZDB-ALT-080523–22 | |

*Continued on next page*

*Continued*

| Reagent type (species) or resource | Designation | Source or reference | Identifiers | Additional information |
|---|---|---|---|---|
| Genetic reagent (zebrafish) | *Tg(foxj1a:gCaMP6s)*[nw9] | This study | N/A | Trangenic zebrafish line expressing the calcium indicator GCamp6s in multiciliated cells of the nose, Jurisch-Yaksi lab, NTNU |
| Genetic reagent (zebrafish) | *Tg(Ubi:zebrabow)* | *Pan et al., 2013* | ZDB-ALT-130816–1 | |
| Genetic reagent (zebrafish) | *mitfa*[b692] | *Lister et al., 1999* | ZDB-ALT-010919–2 | |
| Antibody | Mouse monoclonal glutamylated tubulin (GT335) | Adipogen | Cat#AG-20B-0020-C100; RRID: AB_2490210 | Dilution 1:400 |
| Antibody | Rabbit polyclonal Gamma-tubulin | Thermo Fisher | Cat# PA5-34815; RRID: AB_2552167 | Dilution 1:400 |
| Antibody | Rabbit Polyclonal anti beta-catenin | Cell Signalling Technologies | Cat#9562; RRID:AB_331149 | Dilution 1:200 |
| Antibody | Chicken Polyclonal Anti-GFP | Abcam | Cat#ab13970; RRID:AB_300798 | Dilution 1:1,000 |
| Antibody | Goat Polyclonal anti-rabbit IgG (H+L) Highly Cross-adsorbed Alexa Fluor 555 | Thermo Fisher | Cat# A32732; RRID:AB_2633281 | Dilution 1:1,000 |
| Antibody | Goat Polyclonal anti-mouse IgG (H+L) Highly Cross-adsorbed Alexa Fluor 647 | Thermo Fisher | Cat#A32728; RRID:AB_2633277 | Dilution 1:1,000 |
| Chemical compound, drug | Alpha-bungarotoxin | Invitrogen | Cat#BI601 | |
| Chemical compound, drug | Ultrapure LMP agarose | Fisher Scientific | Cat#16520100 | |
| Chemical compound, drug | DAPI | Invitrogen | Cat# D1306 | Dilution 1:1,000 |
| Software, algorithm | ImageJ/Fiji | *Schindelin et al., 2012* | | |
| Software, algorithm | Cell counter plugin for Fiji/ImageJ | Kurt De Vos, University of Sheffield | https://imagej.net/Cell_Counter | |
| Software, algorithm | BigWarp | Saalfeld lab, Janelia https://imagej.net/BigWarp; *Bogovic et al., 2016* | | |
| Software, algorithm | Zebrascope software in Labview | Ahrens lab, Janelia Farm; *Vladimirov et al., 2014* | | |
| Software, algorithm | Manta Controller | Yaksi lab, NTNU; *Reiten et al., 2017* | | |
| Software, algorithm | Fast Fourier Analysis | MATLAB, this paper; *Jurisch-Yaksi, 2023* | https://github.com/Jurisch-Yaksi-lab/CiliaCoordination | |
| Software, algorithm | Coherence analysis | MATLAB, this paper; *Jurisch-Yaksi, 2023* | https://github.com/Jurisch-Yaksi-lab/CiliaCoordination | |
| Software, algorithm | Wave analysis | MATLAB, this paper; *Jurisch-Yaksi, 2023* | https://github.com/Jurisch-Yaksi-lab/CiliaCoordination | |

*Continued on next page*

*Continued*

| Reagent type (species) or resource | Designation | Source or reference | Identifiers | Additional information |
|---|---|---|---|---|
| Software, algorithm | Computation model of cilia carpet | *Solovev and Friedrich, 2022b*; *Solovev and Friedrich, 2021a*; *Solovev and Friedrich, 2021b*; *Solovev and Friedrich, 2021c* | https://github.com/icemtel/reconstruct3d_opt, https://github.com/icemtel/stokes, and https://github.com/icemtel/carpet | |
| Software, algorithm | ColorBrewer: Attractive and Distinctive Colormaps | *Brewer, 2022*; Cynthia Brewer | https://github.com/DrosteEffect/BrewerMap/releases/tag/3.2.3, GitHub. Retrieved December 4, 2022 | |
| Software, algorithm | Beeswarm | *Stevenson, 2019*; Ian Stevenson | https://github.com/ihstevenson/beeswarm GitHub. Retrieved December 4, 2022. | |
| Other | Sutter laser puller | Sutter | Model P-200 | pulling needles for injection |
| Other | Pressure injector | Eppendorf | Femtojet 4i | injection of bungartoxin for paralysis |
| Other | Confocal microscope | Zeiss | Examiner Z1 | confocal imaging |
| Other | 20 x water immersion Plan-Apochromat NA 1 | Zeiss | 421452-9880-000 | confocal imaging |
| Other | Light-sheet objective | Nikon | 20 x Plan-Apochromat, NA 0.8 | light-sheet imaging |
| Other | Transmission microscope | Bresser, Olympus | | transmission imaging |
| Other | Transmission microscope objective | Zeiss | 63 X, NA 0.9 | transmission imaging |

## Experimental model and subject details

The animal facilities and maintenance of the zebrafish, *Danio rerio*, were approved by the NFSA (Norwegian Food Safety Authority). Fish were kept in 3.5 L tanks in a Techniplast Zebtech Multilinking system at constant conditions: 28 °C, pH7 and 600μSiemens, at a 14:10 hr light/dark cycle to simulate optimal natural breeding conditions. Fish were fed twice a day with dry food (ZEBRAFEED; SPAROS I&D Nutrition in Aquaculture), and once with *Artemia nauplii* (Grade0, platinum Label, Argent Laboratories, Redmond, USA). Larvae were maintained in egg water (1.2 g marine salt and 0.1% methylene blue in 20 L RO water) from fertilization to 3 dpf and subsequently in AFW (1.2 g marine salt in 20 L RO water). For experiments, the following fish lines were used: *Et(hspGGFF19B:Gal4)Tg(UAS:gfp)* (*Reiten et al., 2017*), *Tg(ubi:zebrabow)*(*Pan et al., 2013*) . *Tg(foxj1a:GCaMP6s)*[nw9] transgenic animals were generated in our laboratory upon co-injection of tol2 transposase mRNA and pCS2-Foxj1a:GCaMP6s plasmid containing the 0.6 kb Foxj1a promoter described in upstream of GCamp6s open reading frame (*Caron et al., 2012*). Experiments were performed on embryos of AB Background. All procedures were performed on zebrafish larvae at 4 dpf in accordance with the directive 2010/63/EU of the European Parliament and the Council of the European Union and the Norwegian Food Safety Authorities.

## Immunohistochemistry and confocal microscopy

Euthanized larvae were fixed in a solution containing 4% paraformaldehyde solution (PFA), 0.3% Triton X-100 in PBS (0.3% PBSTx) for at least 3 hr at room temperature. Larvae were washed with 0.3% PBSTx (3x5 min) and permeabilized with acetone (100% acetone, 10 min incubation at −20 °C). Subsequently, samples were washed with 0.3% PBSTx (3x10 min) and blocked in 0.1% BSA/0.3% PBSTx for 2 hr. Larvae were incubated with glutamylated tubulin (GT335, 1:400, Adipogen), and beta-catenin (#9562, 1:200, Cell Signalling Technologies) or gamma-tubulin (1:400, Thermo Scientific) overnight

at 4 °C. On the second day samples were washed (0.3% PBSTx, 3x1 hr) and subsequently incubated with the secondary antibody (Alexa-labelled GAM555 plus, Thermo Scientific, 1:1,000) overnight at 4 °C. The third day, samples were incubated with 0.1% DAPI in 0.3% PBSTx, Life Technology, 2 hr, washed (0.3% PBSTx, 3x1 hr), and transferred to a series of increasing glycerol concentrations (25%, 50%, and 75%). Stained larvae were stored in 75% glycerol at 4 °C and imaged using a Zeiss Examiner Z1 confocal microscope with a 20 x plan NA 0.8 objective. An antibody against GFP (Millipore, ab16901, and Alexa488-anti-chicken, Thermo Scientific) was used to enhance the GFP signal if needed. Acquired images were processed with Fiji/ImageJ (*Schindelin et al., 2012*). Numbers of cells were counted using the "Cell Counter" plugin for Fiji/ImageJ (Kurt De Vos, Univ Sheffield, Academic Neurology, https://imagej.nih.gov/ij/plugins/cell-counter.html). The apical surfaces were collected from the beta-catenin staining using Fiji. The ciliary length was measured from cilia in their extended shape in the *hspGGFF19B;UAS:GFP* line using Fiji. The number of cilia was acquired from the gamma-tubulin staining, at 40 X. Using Fiji, first an ROI was drawn around the apical surface of a cell. The crop was then rescaled and applied with an FFT bandpass filter (up to 50 px; down to 6 px; Suppress stripes: None; Tolerance of direction: 50%), and finally used the detect peaks function in Fiji (prominence = 50). Ciliary direction was identified per cell by comparing the relative position of gamma tubulin compared to glutamylated tubulin, imaged at both 20 X and 40 X. All measurements were further processed in Matlab.

*Tg(ubi:zebrabow)*(*Pan et al., 2013*) larvae were imaged at 20 X. The image was binarized in Fiji, and the surface of the zebrafish was rendered in Matlab.

## Light transmission recordings in the larval nose

High-speed microscopy recordings of motile cilia were conducted with 4-day-old larvae paralyzed with an intramuscular injection of α-bungarotoxin Invitrogen BI601, 1 mg/mL and then embedded in 0.75% low melting point agarose prepared in AFW in a FluoroDish (World Precision Instruments). Specifically, animals were mounted carefully at a 15 degree angle and titled to one side, fully exposing the lateral region of the nose. Agarose covering the nostrils was removed to allow free ciliary beating. The mounted fish was left to rest for 5 min to properly set the agarose and avoid drift. Olfactory pits of zebrafish larvae were visualized by a Bresser transmitted light microscope using a 63 x water immersion objective lens (Zeiss, NA 0.9, plan). The microscope was stabilized using Sorbothane feet blocks (Thorlabs; AV4 - Ø27.0 mm). High-speed digital recordings were captured with an Allied Vision Manta camera (G-031B) at 99–110 frames per second (Fs) at a resolution of 0.15 µm/pixel. Frames were acquired and stored by custom made software written in C++, and further analysis was conducted in MATLAB (Mathworks).

## Motile cilia-mediated fluid flow recordings with fluorescent particles

First a CBF analysis was performed with an Olympus transmitted light microscope, a 63 x water immersion objective lens (Zeiss, NA 0.9, plan) and an Allied Vision Prosilica (GT2000) at a frame rate of 99–110 Hz with a resolution of 0.09375 µm/pixel. Then cilia-mediated flow fields around zebrafish noses were recorded for the same animals, keeping the positioning unchanged. 2 ml of 1.5% fluorescent particle solution in AFW (1.33 µm, Spherotech) was carefully pipetted into the recording chamber. The fish was left to rest for five minutes, letting the particles settle to avoid any drift in the flow. After a 60 s recording, flow fields were generated on MATLAB and the flow direction was determined manually using Fiji.

## Light transmission recordings in the adult brain explant

To measure ciliary beating in the adult zebrafish brain, we used an optimized protocol described earlier (*D'Gama et al., 2021*, *Jeong et al., 2022*). We dissected the brain of nacre (*mitfa*[b692]) adult zebrafish (male and female, less than 1 year-old) in cold artificial cerebrospinal fluid (aCSF). Then we placed the brain explant on a FluoroDish and perfused the brain explants with oxygenated artificial cerebrospinal fluid (aCSF) at room temperature. Ciliary beating of the tela choroidae in the brain explant was recorded using an Olympus transmitted light microscope, a 40 x water immersion objective (Olympus, NA 0.8) and a manta camera (Prosilica GT1930, Allied Vision) at circa 100 Hz with a resolution of 0.314 µm/pixel. Further analyses were performed in MATLAB (Mathworks).

## Light sheet recordings

*hspGGFF19B;UAS:GFP* larvae were imaged using a custom-build light-sheet microscope (based on the design described by *Vladimirov et al., 2014*) with a 20 x water immersion objective (Olympus, NA 1.00, plan), a 4 X illumination objective (Olympus, NA 0.28, XLFLUOR-340), a Hamamatsu CMOS camera (C11440-42U30), PXI system for instrument control (NI PXI 1042Q), and a laser of 488 nm wavelength (Cobolt). Multiciliated cells in the zebrafish nose were recorded for 1–2 s at 300–500 Hz. Paralyzed larvae were mounted in 1.5% Low melting point agarose, agarose was removed in front of their nose, and larvae were positioned in a custom-build chamber so that the multiciliated cells directly faced the light sheet. Images were acquired by the Zebrascope software in LabView (*Vladimirov et al., 2014*) and analysed in ImageJ/Fiji and MATLAB. The light-sheet was adapted to allow for sequential light transmission image acquisition using the same Zebrascope software. Sequential imaging in light-sheet and light transmission mode of *hspGGFF19B;UAS:GFP* animals (4dpf), sparsely labeling multiciliated cells in the nose, to compare the contribution of ciliary beating of one cell to the entire ciliated epithelium. In addition, the light transmission mode was used to acquire sequential ciliary beat recordings with 10 μm spacing (100 Hz), effectively creating a 3D ciliary beating map of the zebrafish nose.

## Viscosity experiments

Methylcellulose 15 cP (M7140-100G; Sigma-Aldrich) was dissolved at 2% (w/v) in Artificial Fish Water (AFW) overnight (500 rpm). Serial dilutions were performed to achieve lower concentrations of methylcellulose in AFW (0.25%, 0.5%, 1%, and 2%), corresponding to (1,875 cP; 3.75 cP; 7.5 cP; 15 cP). Using light transmission microscopy as described above, a 60 s baseline was first recorded at 100 Hz in AFW, after which AFW was exchanged for a viscous solution. After settling for five minutes, the nose is recorded again for 60 s. This was repeated until the fish were recorded with all increasing concentrations and a return to baseline AFW. In another set of experiments, light-sheet experiments were performed as described above. A 4 s baseline was first recorded at 500 Hz in AFW, after which AFW was exchanged for increasingly more viscous solutions. After settling for five minutes, the nose is recorded again for 4 s. Data were further analyzed for CBF, using the framework described above, and ciliary waveforms in Fiji.

## Analysis pipeline for wave direction and wavelength

Beating cilia interfere with the transmission of light and result in an oscillatory change of light intensity over time (*Sanderson and Sleigh, 1981*). We established an analysis pipeline based on the analysis in *Reiten et al., 2017*.

> All codes represented in *Figure 4B–F* are published on Github https://github.com/Jurisch-Yaksi-lab/CiliaCoordination (copy archived at https://doi.org/10.11582/2023.00006).
> A large subset of datasets and their associated codes are published at https://doi.org/10.11582/2023.00006 (*Jurisch-Yaksi, 2023*).

We used the university's servers to run computationally heavier segments of the code – especially *Coherence and coherence-versus-distance calculations*. Since the hardware is shared between users of those servers, available resources are at most 384 GB memory, 2.60 GHz CPU, and Nvidia Tesla K80 GPU.

### Frequency

We analyzed the frequency of oscillations for every pixel of the recording by computing the Fourier transform of the corresponding intensity time series of that pixel using the Matlab fast Fourier transform (*fft()*). To increase computational speed, we spatially down-sampled each frame by a factor 5. For each pixel, cilia beat frequency (CBF) was calculated as the frequency (between 15 Hz and half the frequency of acquisition) where the absolute value of the Fourier transform was maximal. This resulted in a spatial CBF map. As beating cilia do not cover the entire recording area, we automatically segmented this area into signal and noise pixels, using a threshold for the local standard deviation (SD). In brief, we moved a 3x3 kernel across the entire frequency map and any pixel belonging to a 3x3 kernel whose SD was below 3 Hz was considered as signal, whereas pixels belonging only to kernels with an SD above 3 Hz were considered as noise. To remove small signal regions, we listed

all connected pixels in the frequency map using the *bwconncomp*() Matlab function, and removed regions with fewer than 500 pixels. Altogether, with SD-thresholding and by removing small signal regions, we robustly identified those regions of the imaging area with beating cilia.

## Coherence and coherence-versus-distance calculations

The coherence score constitutes a normalized cross-power spectral density: for a pair of time series $x(t)$ and $y(t)$, it is defined as

$$C_{xy}(f) = \frac{|P_{xy}(f)|^2}{P_{xx}(f) P_{yy}(f)},$$ (1)

where

- $C_{xy}(f)$ is the magnitude-squared coherence score of the two time series $x(t)$ and $y(t)$ as function of frequency $f$
- $P_{xy}(f)$ is the cross-power spectral density of $x(t)$ and $y(t)$
- $P_{xx}(f)$ and $P_{yy}(f)$ are the power spectral densities of $x(t)$ and $y(t)$, respectively.

One may think of $P_{xx}(f)$ and $P_{yy}(f)$ as the squared magnitude of the Fourier transforms of $x(t)$ and $y(t)$, whereas $P_{xy}(f) \sim x^*(f) y(f)$ is related to the product of the Fourier transforms of $x(t)$ and $y(t)$.

We apply the coherence score to the intensity time series $x(t)$ and $y(t)$ corresponding to a pair of signal pixels. *Figure 2D* shows an example of input time series and resulting coherence score, see top and bottom panels.

Specifically, coherence scores were calculated for 30 s long raw recordings (approx. 99–110 frames per second; parameter: Fs) using an adapted version of the Matlab *mscohere*() built-in function. In short, *mscohere*() takes two signals and returns the coherence as a function of frequency with values between 0 and 1 ($C_{xy}(f)$). In detail, we first calculated the power spectra for the two time series $x(t)$ and $y(t)$ (Pxx(f)) and (Pyy(f)) using Welch's method with parameters *pwelch(Pxx, window, noverlap, nfft, Fs, 'psd')*. Second, we calculated the cross spectrum for the $x(t)$ and $y(t)$, (Pxy(f)) using the Matlab function *cpsd(Pxx, Pyy, window, noverlap, nfft, Fs)* with parameters window = *hamming*(100); noverlap = 80; nfft = 100. Finally, we estimated a dimensionless coherence score from the power spectra and cross spectrum according to *Equation 1*.

To evaluate coherence in space, we either calculated the coherence for one reference pixel with all other pixels, or for all pixel-pairs in the recording. We use the latter pixel-pairs to plot coherence-*versus*-distance, a measure similar to the correlation-*versus*-distance analysis employed by previous authors (*Bartoszek et al., 2021*; *Jetti et al., 2014*). Since *mscohere*() recalculates the power spectra for a given pixel each time coherence is estimated, we reduced the computing costs by retaining the power spectra. In addition, when calculating the coherence for all pixel-pairs, we spatially down-sampled each frame by a factor 5 and discarded all pixels outside the signal region.

In the end, to visualize the coherence-*versus*-power, and coherence-*versus*-distance measured, we plotted the density using *hexscatter*()(*Bean, 2021*), using distance bins of width 0.5 μm and bins of width 0.04 for the coherence.

## Segmentation

To segment the frequency map into distinct frequency patches, we first bin the power spectra into 100 bins (~0.5 Hz). We then divide the map into frequency patches by grouping same frequency pixels using *bwconncomp*() with two-dimensional eight-connected neighborhood as the desired connectivity, and finally set a minimum size for a patch to 200 pixels. These frequency patches underlie the further analysis.

## Phase

Phase, or phase angle, was extracted from the complex Fourier output by taking the *angle*(). But instead of taking a pixel-based approach, we took a patch-based approach: for all pixels of a given patch, we extracted the phase angle at the main frequency of that patch.

## Gradients

From the segmented phase angle maps, we generated gradient maps by computing the image gradient *imgradientxy(phase, 'prewitt')* over both x and y direction, as well as over phase and phase+ π to adjust for the circular nature of the phase. From these gradient maps, wave directions are extracted by taking the mean angle per patch, while wavelengths are extracted by taking the mode magnitude per patch.

## Detection of ciliary direction and rotating using BigWarp

To compare the noses of different fish, we aligned raw recording frames using BigWarp (*Bogovic et al., 2016*). We choose to perform the *rotation* morphometric transformation. The raw recordings were preselected based on their anatomy: we found that fish exposing much of the lateral region of the nose, tend to have a stronger and larger signal. All animals were aligned to a reference fish left nose (*Figure 4*). Upon aligning the recordings based on spatial features such as, pit edges, centre, and skin regions, we extracted the landmarks before and after the transformation. In addition, all right noses were flipped vertically preceding the transformation. Finally, we applied the transformations, associated with the imported landmarks, to the outputs of the analysis pipeline: frequency, phase, wave direction and wavelength maps.

## Computational model

We consider a computational model of a cilia carpet, where cilia are regularly spaced on a triangular lattice (lattice spacing 18 μm) with aligned cilia polarity, see *Solovev and Friedrich, 2022b* for details. Specifically, we employed a unit cell of 16x16 cilia, for which periodic boundary conditions are imposed. This defines a finite set of possible perfect traveling waves of cilia phase compatible with the boundary conditions, each characterized by different wave direction angle θ and wavelength $\lambda$, see *Figure 6A*. Insets visualize example waves, where the color of dots at respective cilia positions encodes cilia phase. Python packages are available on github to (i) reconstruct 3D curves from orthogonal 2D projections: https://github.com/icemtel/reconstruct3d_opt, (ii) create triangulated surface meshes and solve the Stokes equation of low-Reynolds number hydrodynamics: https://github.com/icemtel/stokes, and (iii) study systems of coupled oscillators with couplings obtained from hydrodynamic computations https://github.com/icemtel/carpet.

## Cilia pumping rate

We computed the mean pumping rate per cilium for different wave solutions as follows: Detailed hydrodynamic computations based on a triangulated surface mesh for a set of neighboring cilia and the boundary surface were conducted using the Stokes equation valid at low Reynolds numbers as detailed in *Solovev and Friedrich, 2022b*. The three-dimensional cilia beat pattern used had been previously tracked from *Paramecium* using stereographic recordings (*Machemer, 1972*). Due to the linearity of the Stokes equation, the individual contributions of all cilia add up and it is sufficient to consider a scenario, where only a single cilium, say at position $\mathbf{x}_j$ and phase $\phi_j$, moves with phase speed $d\phi_j/dt$, while all other cilia at positions $\mathbf{x}_i$ remain immotile with fixed phase $\phi_i$. Detailed hydrodynamic computations then yielded a surface distribution $\mathbf{f}(\mathbf{x})$ of hydrodynamic friction forces, which depends on the vector of cilia phases and is linear in the phase speed $d\phi_j/dt$. Exploiting once more the linearity of the Stokes equation, the instantaneous pumping rate $\mathbf{Q}_j$ of the central cilium can be computed as a surface integral of all local contributions to fluid pumping arising from this force density over the surface $S$ of all cilia. The pumping due to a point-force of magnitude $\mathbf{F}$ coplanar with a planar no-slip boundary at height $z$ above the surface is given by $\mathbf{F}\, z\,/\,(\pi\,\eta)$, where $\eta$ is the dynamic viscosity of the fluid (*Osterman and Vilfan, 2011*). This result was found by integrating Blake's fundamental solution for the flow field induced by a point force near a no-slip boundary; up to prefactor, this result follows also from symmetry considerations and dimensional analysis. We thus find for the instantaneous pumping rate vector (with units of volume pumped per unit time)

$$\mathbf{Q}_j(\phi_1, \cdots, \phi_N) = (\pi\eta)^{-1} \int_S d^2\mathbf{x}\, \mathbf{f}(\mathbf{x})\, z.$$

To economize simulations, this surface integral can be split into contributions from integrating over the surface of different cilia as $\mathbf{Q}_j = \Sigma_I \mathbf{Q}_{ij}$. Importantly, here the contribution $\mathbf{Q}_{ij}$ stemming from integrating over the surface $S_i$ of cilium $i$ depends on the phase $\phi_j$ of the central cilium that is moving and the phase $\phi_i$ of non-moving cilium $i$, but is virtually independent of the phases of the other cilia, i.e., $\mathbf{Q}_{ij} \approx \mathbf{Q}_{ij}(\phi_i,\phi_j)$. Due to the linearity of the Stokes equation, each of these contributions is linear in the phase speed of the central cilium, $\mathbf{Q}_{ij}(\phi_i,\phi_j)=\mathbf{q}_{ij}(\phi_i,\phi_j)$ d$\phi_j$/d$t$ with pumping coefficient $\mathbf{q}_{ij}(\phi_i,\phi_j)$. This allows efficient tabulation of computation results for combinations of neighbors and pairs of cilia phases. Moreover, we confirmed that hydrodynamic interactions between distant cilia are weak and therefore only included the contributions from nearest and certain next-to-nearest neighbors (along the direction of the cilia effective stroke), for which hydrodynamic are strongest (*Solovev and Friedrich, 2022b*). Generally, the contribution from the other cilia, $\mathbf{Q}_{ij}$ for $i \neq j$, points in the direction opposite of $\mathbf{Q}_{jj}$ (but is weaker in magnitude), because the other cilia act as obstacles for the fluid flow generated by the central cilium $j$. Finally, the instantaneous pumping rate vector $\mathbf{Q}_j$ $(\phi_1,...,\phi_N)$ was averaged over a full beat cycle and all cilia indices $j$, to yield a mean pumping rate vector $\mathbf{Q}$ per cilium.

Of note, in an infinite cilia carpet of density $\rho$, the mean pumping rate per cilium sets a slip velocity of approximately steady fluid flow $\mathbf{v}_{slip}$ at a suitable height above the cilia carpet as (*Osterman and Vilfan, 2011*)

$$\mathbf{v}_{slip} = \pi \rho \, \mathbf{Q} = \mathbf{F} \, h/\eta.$$

We computed the pumping rate vector $\mathbf{Q}$ for different traveling waves, where the wave vector $\mathbf{k}$ of the waves determines the phase difference of neighboring cilia. The scalar magnitude $Q=|\mathbf{Q}|$ is reported for all possible $\mathbf{k}$ in *Figure 6B*. There, the scalar pumping rates $Q(\mathbf{k})$ were normalized by the pumping rate $Q_{rand} = |\mathbf{Q}_{rand}|$ for cilia beating with random phase relationship. For the cilia density and beat pattern used, $\mathbf{Q}_{rand} / (\mu m^3/ms) \approx 1.452 \, \mathbf{e}_x+7.739 \, \mathbf{e}_y$, with unit vector $\mathbf{e}_y$ pointing in the direction of the effective stroke of the cilia beat. This pumping rate $\mathbf{Q}_{rand}$ used for normalization was determined by initializing cilia phases with random values (uniformly distributed in the interval [0,2π]), and integrating the dynamics of the cilia carpet over one full beat cycle (corresponding to the increase of the global phase by 2π); results were averaged over n=100 stochastic realizations.

The mean pumping rate vector $\mathbf{Q}$ is parallel to the boundary surface. We can thus report its direction angle relative to the direction of the effective stroke of the cilia beat, see *Figure 6C*.

## Impact of noise

To investigate the effect of imperfect synchronization, we performed dynamic simulations of the cilia carpet model from *Solovev and Friedrich, 2022b* with perturbed initial conditions and computed the mean pumping rate averaged over one beat cycle. Specifically, initial conditions were chosen as a perfect traveling wave as indicated, with independent normal distributed random numbers with variance σ² added to the initial phase of each cilium (as well as a global random phase offset uniformly distributed in the interval [0,2π]). The dynamics was then integrated for a full beat cycle, see *Figure 6D*. For large values of σ, the pumping rates converged to the rate $Q_{rand}$ corresponding to cilia beating with random phase relationship. All results represent averages of n=100 independent realizations.

## Critical density

The dilute spacing of cilia in the computational model prevents steric interactions between cilia. We determined for each possible wave solution with wave vector $\mathbf{k}$ the highest density $\rho_\mathbf{k}$ at which still no steric interactions occur at any point of the beat cycle (determined by a minimal distance of the respective centerlines of neighboring cilia of more than 0.5 μm) by down-scaling the lattice spacing of the triangular arrangement of cilia, see *Figure 6E*. Analogously, we determined a similar critical density $\rho_{rand}$ for cilia beating with random phase relationship, by testing all possible phase relationships of neighboring cilia.

We confirmed that results are robust: if for a pair of neighboring cilia instead of the exact value of the phase difference Δϕ between their respective phases as determined by the wave solutions under consideration, a phase difference from the interval [Δϕ-σ,Δϕ+σ] is chosen, we obtain very similar results even for phase fluctuations with σ=0.4.

## Acknowledgements

We thank V Nguyen, A Nygaard and the Trondheim fish facility team for their technical support, E Yaksi and F Palumbo for the helpful discussions. This work was supported by a Boehringer Ingelheim Fonds fellowship and the Research Council of Norway (RCN) grant 326003 (CR), RCN FRIPRO grants 314189 (NJY). BMF is supported by a DFG-Heisenberg grant (FR3429/4-1), as well as 'Physics of Life' (Cluster of Excellence EXC-2068) and *cfaed*. AS is supported by the DFG priority program SPP1726 'Microswimmers' (FR3429/1-1 and FR3429/1-2 to BMF).

## Additional information

### Funding

| Funder | Grant reference number | Author |
|---|---|---|
| Boehringer Ingelheim Fonds | | Christa Ringers |
| Research Council of Norway | 314189 | Nathalie Jurisch-Yaksi |
| Deutsche Forschungsgemeinschaft | FR3429/1-1 | Benjamin M Friedrich |
| Deutsche Forschungsgemeinschaft | FR3429/1-2 | Benjamin M Friedrich |
| Deutsche Forschungsgemeinschaft | FR3429/4-1 | Benjamin M Friedrich |

The funders had no role in study design, data collection and interpretation, or the decision to submit the work for publication.

### Author contributions

Christa Ringers, Conceptualization, Data curation, Software, Formal analysis, Funding acquisition, Investigation, Visualization, Methodology, Writing – original draft, Writing – review and editing; Stephan Bialonski, Software, Formal analysis, Visualization, Methodology, Writing – review and editing; Mert Ege, Validation, Methodology, Writing – review and editing; Anton Solovev, Software, Formal analysis, Methodology, Writing – review and editing; Jan Niklas Hansen, Software, Formal analysis, Investigation, Methodology, Writing – review and editing; Inyoung Jeong, Investigation, Methodology, Writing – review and editing; Benjamin M Friedrich, Conceptualization, Resources, Data curation, Software, Formal analysis, Supervision, Funding acquisition, Investigation, Visualization, Methodology, Writing – original draft, Writing – review and editing; Nathalie Jurisch-Yaksi, Conceptualization, Data curation, Software, Formal analysis, Supervision, Funding acquisition, Validation, Investigation, Visualization, Methodology, Writing – original draft, Project administration, Writing – review and editing

### Author ORCIDs

Christa Ringers http://orcid.org/0000-0002-0807-8481
Mert Ege http://orcid.org/0000-0002-1498-208X
Jan Niklas Hansen http://orcid.org/0000-0002-0489-7535
Inyoung Jeong http://orcid.org/0000-0003-3554-3032
Benjamin M Friedrich http://orcid.org/0000-0002-9742-6555
Nathalie Jurisch-Yaksi http://orcid.org/0000-0002-8767-6120

### Ethics

The animal facilities and maintenance of the zebrafish, Danio rerio, were approved by the NFSA (Norwegian Food Safety Authority). All procedures were performed on zebrafish larvae at 4 dpf in accordance with the directive 2010/63/EU of the European Parliament and the Council of the European Union and the Norwegian Food Safety Authorities.

### Decision letter and Author response

Decision letter https://doi.org/10.7554/eLife.77701.sa1

Author response https://doi.org/10.7554/eLife.77701.sa2

## Additional files

### Supplementary files
• Transparent reporting form

### Data availability

All codes and a large set of experimental data is published at the following URL https://doi.org/10.11582/2023.00006. Matlab codes for analysis is available on github https://github.com/Jurisch-Yaksi-lab/CiliaCoordination (and archived at https://doi.org/10.11582/2023.00006) Python packages are available on github to (i) reconstruct 3D curves from orthogonal 2D projections: https://github.com/icemtel/reconstruct3d_opt, (ii) create triangulated surface meshes and solve hydrodynamic Stokes equation: https://github.com/icemtel/stokes, and (iii) study systems of coupled oscillators https://github.com/icemtel/carpet.

The following dataset was generated:

| Author(s) | Year | Dataset title | Dataset URL | Database and Identifier |
|---|---|---|---|---|
| Jurisch-Yaksi N | 2023 | Analytical tools to measure cilia synchronization by Ringers et al | https://doi.org/10.11582/2023.00006 | NIRD Research Data Archive, 10.11582/2023.00006 |

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
