## [Editor Report]

This fundamental study reports new observations on the coordination of cilia in zebrafish multiciliated epithelia. The work combines novel experimental methods and computation to provide convincing evidence for a conjectured relationship between local and global synchronization in the form of metachronal waves. The work will be of broad interest to researchers in the areas of cell biology, development, and physiology.

---

## [Decision Letter]

**Decision letter after peer review:**

Thank you for submitting your article "Local synchronization of cilia and tissue-scale cilia alignment are sufficient for global metachronal waves" for consideration by *eLife*. Your article has been reviewed by 2 peer reviewers, and the evaluation has been overseen by a Reviewing Editor and Aleksandra Walczak as the Senior Editor. The reviewers have opted to remain anonymous.

Essential revisions:

1) We take issue with the strong message conveyed by the title – If anything, the study showed that local synchronization of cilia etc is necessary for global metachronal waves – the manuscript does not answer the question of how the waves emerge – there may well be additional mechanisms at play that are not accounted for (indeed the difference between left/right noses strongly hints at some additional mechanisms), so this study merely shows necessity rather than sufficiency. We suggest focusing the title on the system. Whilst the statement in the title ("sufficient") is true, the fact that biological systems might in general behave like this is unknown… conclusions could depend quite strongly on the liquid medium (e.g. mucus or watery), on the 3d geometry which could lead to different sorts of flow recirculation, on the details of cilia arrangements which might be species dependent, etc.

2) The title and the abstract heavily assume that readers know what a metachronal wave is… we doubt that all readers will understand the somewhat subtle difference between the direction and coherence in the wave of coordination and the direction of fluid transport. Maybe the abstract should try to be more introductory in this sense.

3) Page 3, "results". how big is the nose "cup"? We found ourselves worrying about patterns of fluid recirculation. The scales in the maps are not immediately obvious to a reader who is not an expert on this organism. Could a diagram, illustrating this "cup" together with cells and flows be made early in the manuscript? Are there any possible consequences of the tissue not being flat, and perhaps specifically cup shaped?

4) Page 3, "results": numbers are provided with too many significant figures. There's no point giving an error with 3 s.f.

5) Data analysis: In looking at Figure 2G we wonder is this special of pixel 1? How does it look like from other reference points? Can authors make perhaps 16 of these, with equally spaced 4 by 4 reference points?

6) Data analysis: Figure 2I: does panel I change at all if authors consider CBF regions with the highest (and lowest) frequencies, i.e. only work with subregions from C?

7) Caption Figure 2 when discussing sync or no sync in D, say gray-red and gray-blue (you need 2 to sync). How far are these pairs of pixels? illustrate on panel B or C all three positions. In the caption of G', clarify that these are the 3 colors from G.

8) Page 6 line 23, give range of viscosities achieved, even if rough values. (We see them in SM but should be here).

9) Page 9. The differences in the left and right nose are really curious. How symmetric are the left and right nose in terms of the various maps (CBF, coherence, etc)? I don't get a clear sense of this from Figure 5… can something more be presented?

10) Our biggest wish would be to see flow data, on the same experiments… could this be done? Flow visualization is quite easy, especially if tracers can be added e.g. by particle tracking or PIV? As it is, the connection to flow is made, but only by mapping to the simulation model, which has various differences (β, density, fixed waveform, etc). What exactly is the function of motile cilia in the nose? Is flow rate the critical factor, or flow directionality? More discussion would help.

11) The model shows there is some tolerance to phase noise – how does this noise compare with data? Can the model help explain why wave direction seems to be more stable than wavelength?

12) The theory predicts that the flow direction is essentially independent of the direction and wavelength of the waves… so why the distinction between the left/right nose? What is the physiological purpose of the two different wave directions?

Can the authors provide more (i) biological explanations for why this might happen (is there directional transport of some chemical morphogen?), (ii) physical explanations for how an array with identical ciliary orientation can exhibit different wave directions?

13) Have the authors checked carefully if there are any other differences between left and right noses, e.g. in terms of beat frequency, distribution/density of cilia, 3D beat pattern, cilia length etc? Is manual tracking of cilia shape possible say from light-sheet recordings (figure S3 showed this was at least possible in some cases)?

14) The viscosity experiments – when viscosity is increased, the improvement in local coherence is not surprising. The increase is coherence was only slight and restricted mainly to ~20um. The results also didn't pick up any changes in the direction or wavelength of the metachronal waves – so what should be the main take-home message from these experiments?

15) There is some subtlety in the spatial phase mapping technique, e.g. in the segmentation of space into equal-frequency patches, in the treatment of time signals etc – if this technique can be applied successfully to a different system, a more careful discussion of these details and the level of systematic vs real biological noise in the system would be helpful. E.g. What's the optimal choice of patch size? What are the criteria for extracting phases – what is the tolerance in frequency variation across regions? Are 30s recordings ideal for this method?

---

## [Author Response]

Essential revisions:1) We take issue with the strong message conveyed by the title – If anything, the study showed that local synchronization of cilia etc is necessary for global metachronal waves – the manuscript does not answer the question of how the waves emerge – there may well be additional mechanisms at play that are not accounted for (indeed the difference between left/right noses strongly hints at some additional mechanisms), so this study merely shows necessity rather than sufficiency. We suggest focusing the title on the system. Whilst the statement in the title ("sufficient") is true, the fact that biological systems might in general behave like this is unknown… conclusions could depend quite strongly on the liquid medium (e.g. mucus or watery), on the 3d geometry which could lead to different sorts of flow recirculation, on the details of cilia arrangements which might be species dependent, etc.

We agree that we do not answer the question of how the metachronal waves emerge, a question, which in fact is open for several decades now and which is rather difficult to address experimentally. Nevertheless, we hope that our experiments could spur new hypotheses on the emergence of metachronal waves. To facilitate further discovery, we have made some of the experimental data and our codes available on Mendeley and Github.

We also agree that additional mechanisms and/or the 3D geometry of the epithelia tissue may affect metachronal coordination.

Following the suggestions of the reviewers, we have changed to title to *(i)* reflect that is merely a necessity for metachronal waves (while sufficiency is somewhat open), and *(ii)* state our model system; the new title reads:

“Novel analytical tools reveal that local synchronization of cilia coincides with tissue-scale metachronal waves in zebrafish multiciliated epithelia”

2) The title and the abstract heavily assume that readers know what a metachronal wave is… we doubt that all readers will understand the somewhat subtle difference between the direction and coherence in the wave of coordination and the direction of fluid transport. Maybe the abstract should try to be more introductory in this sense.

We thank the reviewers for raising this important point. We revised the abstract to make it more accessible to the broad readership of *eLife*. We also have rephrased the introduction of metachronal waves in the introduction and results part, using a simpler language. Below is the newly edited abstract.

“Motile cilia are hair-like cell extensions that beat periodically to generate fluid flow along various epithelial tissues within the body. In dense multiciliated carpets, cilia were shown to exhibit a remarkable coordination of their beat in the form of traveling metachronal waves, a phenomenon which supposedly enhances fluid transport. Yet, how cilia coordinate their regular beat in multiciliated epithelia to move fluids remains insufficiently understood, particularly due to lack of rigorous quantification. We combine experiments, novel analysis tools, and theory to address this knowledge gap. To investigate collective dynamics of cilia, we studied zebrafish multiciliated epithelia in the nose and the brain. We focused mainly on the zebrafish nose, due to its conserved properties with other ciliated tissues and its superior accessibility for non-invasive imaging. We revealed that cilia are synchronized only locally and that the size of local synchronization domains increases with the viscosity of the surrounding medium. Even though synchronization is local only, we observed global patterns of traveling metachronal waves across the zebrafish multiciliated epithelium. Intriguingly, these global wave direction patterns are conserved across individual fish, but different for left and right nose, unveiling a chiral asymmetry of metachronal coordination. To understand the implications of synchronization for fluid pumping, we used a computational model of a regular array of cilia. We found that local metachronal synchronization prevents steric collisions, cilia colliding with each other, and improves fluid pumping in dense cilia carpets, but hardly affects the direction of fluid flow. In conclusion, we show that local synchronization together with tissue- scale cilia alignment coincide and generate metachronal wave patterns in multiciliated epithelia, which enhance their physiological function of fluid pumping.”

3) Page 3, "results". how big is the nose "cup"? We found ourselves worrying about patterns of fluid recirculation. The scales in the maps are not immediately obvious to a reader who is not an expert on this organism. Could a diagram, illustrating this "cup" together with cells and flows be made early in the manuscript? Are there any possible consequences of the tissue not being flat, and perhaps specifically cup shaped?

Again, we thank the reviewers for raising our awareness for this point. We have now addressed the comments by revising the figures as described below.

In Figure 1, we now include a 3D rendering of the zebrafish snout (former Figure S1A) and a zoom-in of the nasal cavity together with scale-bars. These data were obtained from a live animal expressing a fluorescent indicator in all cells; hence the structure is not altered by fixation.

We have also included a scheme where we drew the location of motile ciliated cells and the fluid flow direction on the 3D rendering of the zebrafish head in Figure 1-supplemental 1 panel A.

As shown in Figure 1A (and mentioned by the reviewers), the nasal cavity is cup-shaped and not flat. Based on the prior literature, such as (Bouderlique et al., 2022; Ferreira et al., 2018; Nawroth et al., 2017), we expect that the 3D geometry has only a small impact on short-range hydrodynamic interactions and local synchronization but can impact the direction of cilia-generated flows. Besides, we focused most of our analysis on the dorsolateral part of the pit because cells are more homogeneously polarized in the dorsolateral part of the pit as compared to the ventromedial part (Figure 5) and the tissue is flatter in the dorsolateral part. Nevertheless, we added the following sentence to the Discussion section to discuss possible impacts of a non-flat geometry: “This difference in wave directions is not a mere consequence of different patterns of cilia alignment, but instead points at an important role of the chirality of the cilia beat and 3D architecture of the underlying tissue”

4) Page 3, "results": numbers are provided with too many significant figures. There's no point giving an error with 3 s.f.

We have now edited the numbers as suggested by the reviewers.

5) Data analysis: In looking at Figure 2G we wonder is this special of pixel 1? How does it look like from other reference points? Can authors make perhaps 16 of these, with equally spaced 4 by 4 reference points?

We thank the reviewer for bringing this point up. We have now included a Supplementary Figure (Figure2- Supplement 3), where we systematically look at 16 reference pixels. The 16 pixels were identified from a 4x4 grid without manual selection and represent well the diversity in coherence score that we observed. Also, as expected, pixels in the non-signal region (no beating cilia present), do not synchronize with other pixels in the nose (these reference pixels are represented by circles instead of crosses).

6) Data analysis: Figure 2I: does panel I change at all if authors consider CBF regions with the highest (and lowest) frequencies, i.e. only work with subregions from C?

Following the suggestion by the reviewer, we tested how the coherence-versus-distance distribution (Figure 2I) changes when selecting CBF regions according to their frequency (analogous to the example shown in Figure 2BC). Specifically, we selected either

– The bottom 33.3% of pixels (red) – low frequencies

– The top 33.3% of pixels (green) – high frequencies

(Note that selecting CBF regions with either the highest or lowest CBF results in only a small number of data points.)

We have not seen any obvious difference for this example case (see Author response image 1). This observation is consistent with our finding that frequency is largely variable across individuals, but the mean coherence and fraction of synchronized pixel remain similar across individuals. Hence, we conclude that frequencies do not directly correlate with synchronization strength.

**Author response image 1. sa2fig1:** Coherence-versus-distance distributions for high and low frequencies. (**A**) Histogram representing the CBF for all pixels and their segmentation into high and low CBF. In red are indicated the bottom 33% CBF values and in green the top 33% CBF values. (**B**) Map showing the location of the high and low CBF. (**C**) The coherence versus distance measure is not majorly affected when analyzing only the top 33% or bottom 33% CBF values.

7) Caption Figure 2 when discussing sync or no sync in D, say gray-red and gray-blue (you need 2 to sync). How far are these pairs of pixels? illustrate on panel B or C all three positions. In the caption of G', clarify that these are the 3 colors from G.

We have now included the location of the pixels of panel D on the map of the ciliated epithelium in panel 2B. The reference pixel is indicated in black, the synchronized example is indicated in red and the non-synchronized example is indicated in blue.

We have also clarified in the figure caption that the colors used in panel G’ correspond to those used in panel G.

8) Page 6 line 23, give range of viscosities achieved, even if rough values. (We see them in SM but should be here).

We have now included this information also in the main text as suggested by the reviewers.

9) Page 9. The differences in the left and right nose are really curious. How symmetric are the left and right nose in terms of the various maps (CBF, coherence, etc)? I don't get a clear sense of this from Figure 5… can something more be presented?

We agree with the reviewers that the differences between the left and right nose are really curious. We have now performed extensive comparison between left and right noses on all quantifiable measures from our experiments. We have not observed any significant differences for median CBF, nose size, nasal cavity size, number of multiciliated cells (MCC) and the directionality of the flow. These data are now included in Figure5 – supplement 3.

We would like to also highlight that there is a large diversity across individual animals (see figure 2 supplement 1 panel A) and our biggest challenge is that we cannot image from both noses of the same fish due to the positioning of the animal. So all data are obtained from a left or a right nose of a different animal, and hence also include the inter-individual differences, which may mask subtle differences between left and right nose.

10) Our biggest wish would be to see flow data, on the same experiments… could this be done? Flow visualization is quite easy, especially if tracers can be added e.g. by particle tracking or PIV? As it is, the connection to flow is made, but only by mapping to the simulation model, which has various differences (β, density, fixed waveform, etc). What exactly is the function of motile cilia in the nose? Is flow rate the critical factor, or flow directionality? More discussion would help.

As prompted by the reviewers, we have now performed CBF and flow measurement sequentially on the same animals. These new data are included in Figure 5F-H and Figure 5 – supplement 4. This new data confirmed the difference in wave direction between the left and the right noses, while net flow directions are approximately mirror symmetric. We observed a consistent wave direction for left noses and a larger variance for the right noses while performing these additional experiments (n=9 right and 10 left). We have now added this new set of data in Figure 5 F-H (green colors for wave directions), updated the quantifications and provided all examples in Figure 5 – supplement 1.

For flow measurement following the CBF measurements, we applied fluorescent beads and measured their movements over a period of 1 minute. We recovered the flow field upon projection of the time series and quantified the direction of the fluid flow at the exit of the nasal cavity, as shown in Figure 5 – supplement 4.

We would like to emphasize that the flow pattern is rather complex due to the 3D geometry of the nose pit and the complex cilia polarity. Hence, we are not able to measure the fluid flow generated solely by the region showing different wave direction, but only the entire output of the ciliated epithelium.

We did not observe any significant difference in the fluid flow direction between the left and right nose as indicated in Figure 5F-G. This provides evidence that the net flow generated by the ciliated epithelium remains identical irrespective of the different wave directions in the left and right nose, which is consistent with the predictions of our computational model.

We had previously published an article on the biological function of cilia-generated fluid flow for olfaction in the zebrafish larvae (Reiten et al., 2017). In this previous work, we showed that flow patterns are important to draw fluid containing odor molecules to the nasal cavity and to expel it afterwards, which improves the temporal resolution of odor detection. So far, we only analyzed the impact of a complete loss of cilia motility, and thus flow, on olfaction, but not yet of only a gradual reduction of flow speed. Notwithstanding, based on our prior work and the literature, we expect that both flow speed and flow direction are important for drawing and expelling odors to the epithelium in an efficient manner.

11) The model shows there is some tolerance to phase noise – how does this noise compare with data?

A direct comparison of phase noise of cilia dynamics in the zebrafish noise and in the computational model is difficult for three reasons, indicated below:

1. Our phase reconstruction algorithm estimates a phase for each pixel, but not for individual cilia (each motile cilium will contribute to several pixels, likewise the intensity of a pixel may comprise contributions from several pixels).

2. Apparent fluctuations in reconstructed phase maps comprise both genuine biological noise of cilia dynamics and measurement noise, which are difficult to separate.

3. The cilia spacing in the zebrafish noise and the computational model are different.

Nonetheless, we can resort to a visual comparison to convery the biological importance of the noise strengths used in the computational model.

For this, we ploted synthetic phase maps perturbed by noise of various strength. Specifically, we assume a phase map of the form

ϕ(x) = k·x + ξ(x),

where **k** = 2π **e** / λ is a wavevector pointing along the direction of the unit vector **e** (enclosing an angle of 10⁰ with the **e***_x_*-axis), λ = 4µm is a typical wavelength, and ξ (**x**) is a phase noise term.

Based on previous theoretical work on a model cilia carpet of hydrodynamically coupled cilia exhibiting active phase noise, we chose the phase noise term such that its Fourier transform modes are stochastically independent and normally distributed with variance that scales inversely with the squared norm of the Fourier mode vector

⟨∅m~|∅m~∗⟩=2D∅ δm,m′ 1|m|2

We refer the interested reader to the Supplemental Material of Solovev *et al.* for additional information and a derivation (Solovev and Friedrich, 2022a). We can choose the noise parameter *D*ϕ such that two phase values at a distance *d*=18µm have a prescribed standard deviation σ.

We have now included visualizations of various noise strength σ of metachronal waves in Figure 6- supplement 1. These data show wave patterns as phase maps perturbed by noise as well as histograms of wave direction computed by using the same image gradient analysis as used for the analysis of experimental data. Even though a direct comparison of phase noise of cilia dynamics in the zebrafish noise and in the computational model is difficult, these results indicate that phase noise does not perturb our analysis of wave direction.

Can the model help explain why wave direction seems to be more stable than wavelength?

Our computational model focuses on fluid pumping. Hence, we would have to resort to more detailed computational models that address synchronization by hydrodynamic interactions to address this question. The video still in Author response image 2 is taken from the Supplemental Material of (Solovev and Friedrich, 2022b) and shows a snapshot of metachronal coordination in a model cilia carpet with active noise, where each colored dot represents cilia phase of a model cilium at the respective position. Wave fronts of metachronal waves are clearly visible, corresponding to a well-defined direction of metachronal waves, whereas relative phase differences between neighboring cilia vary more. We assume that fluctuations in metachronal wave direction would result in hydrodynamic interactions favoring incompatible phase differences between neighboring cilia (and possibly steric collisions in dense carpets). This will in turn attenuate as compared to variations in wavelength, which will arise already from small variations in the relative phase difference between neighboring cilia.

This qualitative argument thus suggests that wave direction should be more stable than wavelength. Future theoretical work should investigate this interesting point more thoroughly.

12) The theory predicts that the flow direction is essentially independent of the direction and wavelength of the waves… so why the distinction between the left/right nose? What is the physiological purpose of the two different wave directions?Can the authors provide more (i) biological explanations for why this might happen (is there directional transport of some chemical morphogen?), (ii) physical explanations for how an array with identical ciliary orientation can exhibit different wave directions?

Computational studies suggest that the direction of metachronal waves direction depends crucially on the shape of the cilia beat pattern (Kanale, Ling, Guo, Fürthauer, and Kanso, 2022; Meng, Bennett, Uchida, and Golestanian, 2021). In fact, our own work and that of others showed that three-dimensional, chiral beat patterns of cilia give rise to anisotropic hydrodynamic interactions between neighboring cilia, which select the direction of emergent metachronal waves (Solovev and Friedrich, 2022a). We speculate that subtle differences in the cilia beat pattern between left and right nose could be the cause for the observed differences in metachronal wave direction. We assume that changes in wave direction would result in frustrated hydrodynamic interactions between neighboring cilia (and possibly steric collisions in dense cilia carpets) and are thus less pronounced that variations of wavelength, which result already from small changes in the relative phase difference between neighboring cilia.

Based on the literature, including our own work, we showed that metachronal wave direction depends on the shape of the beat pattern (Meng et al., 2021; Solovev and Friedrich, 2022b).

Given that we have not observed any differences between the left and the right nose for the anatomy, CBF and overall flow direction, but only for the metachronal wave direction, it is possible that this difference may reflect an epiphenomenon with no direct biological function.

13) Have the authors checked carefully if there are any other differences between left and right noses, e.g. in terms of beat frequency, distribution/density of cilia, 3D beat pattern, cilia length etc? Is manual tracking of cilia shape possible say from light-sheet recordings (figure S3 showed this was at least possible in some cases)?

As mentioned above in point 9, we have not observed any statistically significant difference between cells in the left and in the right noses. It would be wonderful to compare cilia beat waveform from various cells taken from the right and the left nose. Tracing methods are currently not suitable for making a conclusive statement as small differences in cilia waveform will be missed. Moreover, individual cells show high level of diversity, which may further mask differences between left and right nose.

High-precision tracking of cilia waveform is well established only for single cilia and planar beat patterns (Geyer, Howard, and Sartori, 2022), but remains a challenge for three-dimensional beat patterns and cilia in dense cilia carpets.

We trust that future technological advances will allow high-precision tracking also for individual cilia on multiciliated cells, thus allowing to address these questions in the future.

14) The viscosity experiments – when viscosity is increased, the improvement in local coherence is not surprising. The increase is coherence was only slight and restricted mainly to ~20um. The results also didn't pick up any changes in the direction or wavelength of the metachronal waves – so what should be the main take-home message from these experiments?

We thank the reviewers for this comment.

We manipulated and quantified ciliary synchronization of an in vivo ciliary carpet to experimentally address the impact of viscosity on synchronization. However, we find that changing viscosity does not matter much. The limited increase in synchronization is consistent with predictions from simple theories of hydrodynamically coupled cilia. (The physical reason behind is that a change in fluid viscosity re-scales hydrodynamic forces but does not change the relative magnitude of hydrodynamic interactions.)

Any effect of viscosity on cilia synchronization should be indirect due to a slight change in cilia beat pattern by the changed viscosity (Brokaw, 1966). Unfortunately, we could not observe these changes directly due to the difficulty of measuring waveforms of individual cilia on multiciliated cells.

Notably, there are important differences between a ciliated epithelium in zebrafish exposed to high- viscosity fluid, and ciliated epithelium in mammalian airways transporting mucus. Still, our result in the viscosity experiments adds support to the generalizability of our model system.

15) There is some subtlety in the spatial phase mapping technique, e.g. in the segmentation of space into equal-frequency patches, in the treatment of time signals etc – if this technique can be applied successfully to a different system, a more careful discussion of these details and the level of systematic vs real biological noise in the system would be helpful. E.g. What's the optimal choice of patch size? What are the criteria for extracting phases – what is the tolerance in frequency variation across regions? Are 30s recordings ideal for this method?

We thank the reviewer for acknowledging the complexity of our spatial phase mapping approach.

While we indeed optimized the parameters of our algorithm for our data, we emphasize that our algorithm is reasonably robust and, based on our experiences, would also give satisfactory results if these parameters were changed. See below for further details.

To highlight the robustness of our methods, we have run our spatial phase map analysis also on another multiciliated tissue, i.e., the ependymal layer of the tela choroida in the zebrafish brain. Results for the tela choroida are now included in Figure 2 – supplement 4 for the coherence data and Figure 4 – supplement 2 for the metachronal wave directions. These results are discussed further below in the point 15.3.

Below we provide a detailed explanation of the choice of algorithm parameters.We invite the reviewers and future users to interactively explore our code, which is available on GitHub (https://github.com/Jurisch-Yaksi-lab/CiliaCoordination) and which automatically generates plots from our deposited data ( https://doi.org/10.11582/2023.00006). We have included all information below in a README file on github. We deposited all our codes and data on public repositories upon acceptance of our manuscript (URL https://doi.org/10.11582/2023.00006).

1. How are frequency maps segmented in space into equal-frequency patches?

We realized early on that the power spectrum has a higher frequency resolution than necessary (0.1Hz) and that small variations in frequency could represent noise rather than cilia beating at different frequencies. Hence, we decided to bin the power spectrum (using a bin width of 0.54Hz).

The precise choice of bin widths had only a small impact on the CBF segmentation (note that the spatial distribution of CBF remain similar). For bin widths grossly outside of this range, too few or too many CBF values result (see Author response image 3 for a visual example).

Based on our experiments with the multiciliated epithelium in the nose and brain of the zebrafish we would recommend using a frequency binning of circa 0.5Hz

**Author response image 3. sa2fig3:** Impact of the binning of the power spectrum on CBF values shown on CBF heatmaps (top) and histograms (bottom). We used a binning of 0.54Hz for all our analysis due to their minimal impact and good coverage of CBF values.

2. How are time signals treated?

Time signals were not altered by our analysis. Binning was performed in the frequency domain.

The only threshold applied in our analysis was for the CBF analysis. CBF were identified between 15Hz and half of the frequency of acquisition (Fs = 100Hz; Nyquist (1/2Fs) = 50Hz), based on the Nyquist formula. We used 15Hz as a threshold for the CBF based on the analysis of n=130 fish nasal epithelia (Reiten et al., 2017; shown in Author response image 4), which identified 15Hz as the lowest frequency in the nasal multiciliated epithelium.

**Author response image 4. sa2fig4:** Histogram showing the average CBF of multiciliated cells in the nose of the zebrafish larvae (n=130 animals). From Reiten et al., 2017.

This value needs to be adapted by future users to the tissue of interest based on their CBF. For instance, a cut-off of 10Hz was used for the ependymal cells of the zebrafish brain based on our earlier work (D'Gama et al., 2021).

We recognize that by imaging with a frame rate of approximately 100Hz, we miss out on higher frequencies. That is a technical limitation of the setup, which allowed us to image the entire nose rather than few cells with higher frequencies of acquisition. Nevertheless, selected control experiments with higher frequency of acquisition and smaller field of view (data not shown), revealed similar findings.

3. How do the subtleties compare in a different system?

We have now performed additional experiments on a different multiciliated tissue, i.e., the ependymal cells of the adult zebrafish brain (shown now in Figure 4-supplement 2). We have previously shown that the ependymal tissue is more sparsely populated by ciliated cells as compared to the nasal epithelium (D'Gama et al., 2021; Jeong, Hansen, Wachten, and Jurisch-Yaksi, 2022). Moreover, ciliated cells have different properties with regard to their CBF, apical size and numbers of cilia. Hence, we needed to optimize parameters for this particular tissue and we recommend experimenters to adapt these values for each ciliated system. Similarly, these parameters should be adapted to each acquisition system, and reflect the spatial resolution of the recordings. As a note, the experiments on the zebrafish brain and the new set of experiments on the zebrafish nose including fluid flow measurements were performed on a different microscope with a different camera. Since the results are highly comparable, we argue that our analysis tools are robust and can easily be adapted across recording setups and ciliated tissues.

4. What is the optimal choice of patch-size?

For quantification of wave direction and wavelength, we opt for frequency patches that are as large as possible but display sufficiently homogeneous CBF.

To this end, we choose the patch-size by:

1. Binning the frequencies (as described above)

2. Setting a minimum patch size of 400 pixels, which corresponds to circa 9µm^2^, roughly corresponding to the area swept over by one cilium. We have also used higher minimum patch size, eg 800pixel but as shown in Author response image 5, this was too stringent.

**Author response image 5. sa2fig5:** Segmentations into frequency patches with or without binning and with a minimum patch size of 400 or 800 pixels. Note that a binning of 0.54Hz and minimum size of 400 pixels (9 µm2) provides the best segmentation of the ciliated epithelium with a reasonable number of patches.

5. What are the criteria for extracting phases? – What is the tolerance in frequency variation across regions?

Phases were extracted from the Fourier transform of the intensity time series of each pixel evaluated at the binned frequency of the segmented patch of that pixel. Using the same evaluation frequency for each pixel in a frequency patch is important to ensure that extracted phases are comparable across the patch. We tested this algorithm extensively on synthetic data of simulated coupled/uncoupled noisy phase oscillators and confirmed that we can robustly estimate their phase from time-series of finite length.

We note a trade-off between precision and accuracy: using longer time series will increase the precision of phase estimation at the expense of reduced temporal resolution and accuracy.

We additionally tested an alternative algorithm to extract cilia phase based on the Hilbert transform of intensity time series which gave results that were consistent with the Fourier-based algorithm (but were not chosen for the present manuscript as results from this alternative algorithm were slightly noisier and less easy to interpret).

6. Are 30s recordings ideal for this method?

For consistent results, the same duration of time series should be used for extracting phases and calculating coherence scores.

The choice of optimal time duration of time series for the analyses represents a trade-off between increasing precision (longer duration of time series) versus accuracy (shorter time series). As a rule of thumb, time series should comprise a sufficient number of oscillation cycles (~500 in our case). For very long time series, additional post-processing steps can become necessary to reduce drift.

As shown in Author response image 6, using a time duration of 10-20s instead of 30s gave consistent results for the coherence score but background values were higher. Conversely, using a longer time duration of up to 240s did not improve the results for the coherence score. As longer recording time require a perfectly stable recording with zero drift which is very difficult to achieve, we recommend the users to select a time window of 30 sec for the coherence score to minimize background and impact of potential drift.

**Author response image 6. sa2fig6:** Coherence analysis for 3 reference pixels for different recording lengths (10-240s). Note that increasing recording length reduces the background values, but do not changes the overall coherence patterns. We recommend a duration of 30s to increase signal-to-noise ratio of the coherence score.

References

Bouderlique, T., Petersen, J., Faure, L., Abed-Navandi, D., Bouchnita, A., Mueller, B.,... Adameyko, I. (2022). Surface flow for colonial integration in reef-building corals. *Curr Biol, 32*(12), 2596- 2609.e2597. doi:10.1016/j.cub.2022.04.054

Brokaw, C. J. (1966). Effects of increased viscosity on the movements of some invertebrate spermatozoa. *J Exp Biol, 45*(1), 113-139. doi:10.1242/jeb.45.1.113

D'Gama, P. P., Qiu, T., Cosacak, M. I., Rayamajhi, D., Konac, A., Hansen, J. N.,... Jurisch-Yaksi, N. (2021). Diversity and function of motile ciliated cell types within ependymal lineages of the zebrafish brain. *Cell Rep, 37*(1), 109775. doi:10.1016/j.celrep.2021.109775

Ferreira, R. R., Pakula, G., Klaeyle, L., Fukui, H., Vilfan, A., Supatto, W., and Vermot, J. (2018). Chiral Cilia Orientation in the Left-Right Organizer. *Cell Rep, 25*(8), 2008-2016.e2004. doi:10.1016/j.celrep.2018.10.069

Jeong, I., Hansen, J. N., Wachten, D., and Jurisch-Yaksi, N. (2022). Measurement of ciliary beating and fluid flow in the zebrafish adult telencephalon. *STAR Protoc, 3*(3), 101542. doi:10.1016/j.xpro.2022.101542

Kanale, A. V., Ling, F., Guo, H., Fürthauer, S., and Kanso, E. (2022). Spontaneous phase coordination and fluid pumping in model ciliary carpets. *Proceedings of the National Academy of Sciences, 119*(45), e2214413119. doi:doi:10.1073/pnas.2214413119

Meng, F., Bennett, R. R., Uchida, N., and Golestanian, R. (2021). Conditions for metachronal coordination in arrays of model cilia. *Proceedings of the National Academy of Sciences, 118*(32), e2102828118. doi:10.1073/pnas.2102828118

Nawroth, J. C., Guo, H., Koch, E., Heath-Heckman, E. A. C., Hermanson, J. C., Ruby, E. G.,... McFall-Ngai, M. (2017). Motile cilia create fluid-mechanical microhabitats for the active recruitment of the host microbiome. *Proceedings of the National Academy of Sciences, 114*(36), 9510-9516. doi:10.1073/pnas.1706926114

Reiten, I., Uslu, F. E., Fore, S., Pelgrims, R., Ringers, C., Diaz Verdugo, C.,... Jurisch-Yaksi, N. (2017).Motile-Cilia-Mediated Flow Improves Sensitivity and Temporal Resolution of Olfactory Computations. *Current Biology, 27*(2), 166-174. doi:https://doi.org/10.1016/j.cub.2016.11.036

Solovev, A., and Friedrich, B. M. (2022a). Synchronization in cilia carpets and the Kuramoto model with local coupling: Breakup of global synchronization in the presence of noise. *Chaos: An Interdisciplinary Journal of Nonlinear Science, 32*(1), 013124. doi:10.1063/5.0075095

Solovev, A., and Friedrich, B. M. (2022b). Synchronization in cilia carpets: multiple metachronal waves are stable, but one wave dominates. *New Journal of Physics, 24*(1), 013015. doi:10.1088/1367- 2630/ac2ae4